# Storm-Resolving Models Advance Atmospheric Blocking Simulations and Climate Change Insights

Edgar Dolores-Tesillos<sup>1,3</sup>, Olivia Martius<sup>1</sup>, and Stephan Pfahl<sup>2</sup>

Correspondence: Edgar Dolores-Tesillos (edgar.dolorestesillos@unil.ch)

#### Abstract.

Atmospheric blocking is a key driver of midlatitude weather extremes, including heatwaves and cold spells. Yet general circulation models (GCMs) still struggle to capture the frequency, persistence, and spatial characteristics of blocking. Here, we evaluate atmospheric blocking in next-generation storm-resolving Earth system models from the nextGEMS, EERIE, and DestinE projects, focusing on ICON and IFS-FESOM with  $\sim 10$  km atmospheric and  $\sim 5$  km ocean grid spacing. We also provide first insights into the IFS-FESOM under SSP3-7.0 forcing.

Blocking frequency, duration, and size are assessed in historical simulations spanning 30 years for IFS and 27 years for ICON, relative to ERA5 reanalysis and a CMIP6 multi-model ensemble of eight models. We further examine links between blocking biases and the background flow, sea surface temperatures (SSTs), and storm-tracks. Performance varies regionally and seasonally: IFS, particularly in its atmosphere-only configuration, reproduces blocking frequency and jet structure more realistically than coupled IFS and ICON over the North Atlantic and North Pacific. ICON shows larger winter biases, including overly zonal jets and underestimated Euro-Atlantic blocking compared to IFS. Several biases identified in the CMIP6 models persist in the storm-resolving models or are even amplified, showing that higher resolution alone does not consistently result in better blocking representation. Atmosphere-only experiments (IFS AMIP) highlight the strong influence of sea surface temperatures (SSTs) and the sensitivity of blocking to ocean–atmosphere coupling. We find a positive relationship between blocking frequency and storm-track activity in JJA in the CMIP6 models, which is weaker or absent in the storm-resolving models.

Under SSP3-7.0, IFS projects reduced winter blocking at high latitudes (e.g., northern Europe) and reduced summer blocking frequency over the North Atlantic, northern Europe, and Russia. Changes in magnitude, spatial pattern, and persistence are often of the same order as the model biases, indicating that projected blocking responses are difficult to disentangle from systematic errors related to jet structure, SST biases, and storm-track activity. Overall, storm-resolving models show local improvements in blocking representation, particularly when forced with realistic SSTs. However, coupled simulations still exhibit large biases, underlining the need for further development of ocean–atmosphere coupling representation. These findings highlight both the potential and the current limitations of storm-resolving models for simulating and projecting persistent weather extremes in a warming climate.

1

<sup>&</sup>lt;sup>1</sup>Institute of Geography, Oeschger Centre for Climate Change Research, University of Bern, Bern, Switzerland

<sup>&</sup>lt;sup>2</sup>Institute of Meteorology, Freie Universität Berlin, Berlin, Germany

<sup>&</sup>lt;sup>3</sup>Faculty of Geosciences and Environment, University of Lausanne, Lausanne, Switzerland

45

55

#### 1 Introduction

Extratropical cyclones and atmospheric blocking are key drivers of midlatitude weather variability. While extratropical cyclones are fast-moving low-pressure systems that bring stormy conditions, atmospheric blocking refers to quasi-stationary, long-lived high-pressure systems that disrupt the typical west-to-east movement of weather systems (Elliott and Smith, 1949; Rex, 1950; Namias, 1964). By persistently deflecting storm-tracks, blocking can maintain extreme conditions such as prolonged heatwaves in summer or cold spells in winter (e.g., Kautz et al., 2022). These events have severe socio-economic impacts, affecting sectors such as energy, agriculture, and public health, making their accurate simulation a priority for climate modeling (e.g., Planchon et al., 2015; Grams et al., 2017; Ormanova et al., 2020; Ackerley et al., 2025).

However, CMIP5 and CMIP6 general circulation models (GCMs) struggle to reproduce key characteristics of atmospheric blocking, including its frequency, location, and duration (e.g., Schiemann et al., 2017, 2020). CMIP6 models, in particular, continue to underestimate blocking frequency in critical regions such as the Euro-Atlantic sector (Dolores-Tesillos et al., 2025). These biases degrade the skill of climate projections, especially regarding the persistence of high-impact weather regimes (Davini and D'Andrea, 2016). Several factors are thought to contribute to these deficiencies, including coarse horizontal resolution, inadequate storm-track representation, and simplified parameterizations of moist diabatic processes (Woollings et al., 2018; Dolores-Tesillos et al., 2025).

The correct representation of the jet waveguide in models is crucial for capturing blocking dynamics. The jet waveguide modulates the propagation of wave packets and the formation of blocking (Nakamura and Huang, 2018). Blocking occurs preferentially in areas where the waveguide strength decreases and the flow becomes diffluent (Nakamura and Huang, 2018; Shutts, 1983).

Sea surface temperature (SST) biases have also been identified as a major contributor to blocking frequency biases (Scaife et al., 2011). For example, a persistent cold SST bias south of Greenland strengthens the low-level meridional temperature gradient, enhancing baroclinicity and the eddy-driven jet, thereby favoring a more zonal flow configuration that suppresses blocking (Scaife et al., 2011; Athanasiadis et al., 2022; Cheung et al., 2023). However, the impact of SST biases on blocking is multifaceted: changes in SST gradients affect low-level baroclinicity via thermal wind balance, while absolute SST anomalies modify latent and sensible heat fluxes, influencing static stability and the moisture availability required for diabatic processes (e.g., Hermoso et al., 2024; Wills et al., 2024). Blocking sensitivity to tropical SST anomalies has also been documented (e.g., Hinton et al., 2009). Ocean–atmosphere feedbacks often emerge on longer timescales. For example, weakened surface winds during blocking events can reduce ocean gyre strength and heat exchange, potentially reinforcing the warm phase of the Atlantic Multidecadal Variability (AMV) (Häkkinen et al., 2011).

In addition to oceanic influences, moist diabatic processes such as latent heat release associated with cloud formation in warm conveyor belts are increasingly recognized as essential for blocking onset and maintenance (Steinfeld and Pfahl, 2019; Steinfeld et al., 2020; Dolores-Tesillos et al., 2025). Their role in midlatitude dynamics is expected to grow in a warming climate (e.g., Dolores-Tesillos et al., 2022; Steinfeld et al., 2022), yet they remain poorly represented in most current global climate models due to resolution and parameterization limitations.

Recent studies have shown that increasing horizontal resolution can improve storm-track and blocking representation by better capturing orography, jet structure, warm conveyor belts, and transient eddies (Berckmans et al., 2013; Willison et al., 2013; Schemm, 2023; De Luca et al., 2024). However, the interactions between resolution, oceanic forcing, dry dynamical processes (e.g., baroclinic instability and waveguides), and moist processes remain poorly understood. This study investigates atmospheric blocking in the Northern Hemisphere using storm-resolving simulations from the nextGEMS, EERIE, and DestinE projects. We analyze historical and SSP3-7.0 simulations with the Integrated Forecasting System coupled to the Finite-volumE Sea ice—Ocean Model (IFS-FESOM) and the ICOsahedral Non-hydrostatic model (ICON), comparing them against ERA5 and CMIP6 models. We assess the influence of model resolution, ocean forcing, and large-scale dynamics on blocking characteristics during both winter and summer.

We aim to answer the following research questions:

- How does horizontal resolution influence the simulation of atmospheric blocking frequency, duration, and spatial structure in storm-resolving models compared to traditional CMIP6 models?
  - What are the key drivers of biases in blocking representation, including the roles of background flow characteristics, storm-track activity, and sea surface temperatures?
  - How do projected changes in blocking frequency and properties under a future warming scenario (SSP3-7.0) compare across storm-resolving simulations?
  - What added value do storm-resolving simulations offer for improving the understanding and projection of blocking behavior in a changing climate?

This manuscript is organized as follows. Section 2 describes the data sources and simulation setup. Section 3 presents the methodology used for identifying and characterizing blocking events and capturing storm-track activity. The main results are presented in Section 4, followed by a discussion of the key findings in Section 5. Finally, Section 6 summarizes the conclusions and outlines future research directions.

#### 2 Data

75

The main data source is the nextGEMS project (Segura et al., 2025), which has developed a new generation of global storm-resolving Earth system models. The first modeling cycle included simulations with the ICOsahedral Non-hydrostatic model (ICON) and the Integrated Forecasting System coupled to the Finite-VolumE Sea ice—Ocean Model (IFS-FESOM), as described by Hohenegger et al. (2023) and Rackow et al. (2025), respectively. All simulation data are publicly available via the World Data Center for Climate (Koldunov et al., 2023; Wieners et al., 2024). Table 1 provides an overview of all simulations analyzed in this study, which are primarily derived from nextGEMS efforts.

The nextGEMS models, ICON and IFS-FESOM, are fully coupled models (atmosphere, ocean, sea ice, and land) at kilometrescale resolution with an energetically consistent climate (Segura et al., 2025). Although capable of including additional com-

100

ponents such as the carbon cycle and aerosols, the simulations used in this study were performed without these Earth system extensions. We refer to ICON and IFS-FESOM as storm-resolving Earth system models. Initial analyses show promising results for climate extremes and mesoscale weather (e.g., Brunner et al., 2025; Poujol et al., 2025; Wille et al., 2025).

A key distinction between the models lies in their treatment of moist convection. In ICON, deep convection is explicitly resolved, with the deep convection parameterization switched off. In contrast, IFS-FESOM continues to use a semi-parameterized approach to convection. Deep, shallow, and mid-level convection are represented using a mass-flux scheme in which the cloud-base mass flux is derived from a modified Convective Available Potential Energy (CAPE) closure (Rackow et al., 2025).

In addition to the nextGEMS simulations, we analyze an atmosphere-only IFS simulation (IFS AMIP; Aengenheyster et al., 2025) from the European Eddy-Rich Earth-System Models (EERIE) project ([https://eerie-project.eu/](https://eerie-project.eu/]). The EERIE project aims to assess the role of ocean mesoscale processes in modulating climate variability from seasonal to centennial scales. The IFS AMIP simulation uses the same atmospheric configuration as the IFS-FESOM setup but is forced with prescribed sea surface temperatures (SSTs) and sea ice concentrations (SICs) from ESA-CCI v3 (Embury et al., 2024). This configuration is particularly useful for isolating the impact of air—sea coupling on large-scale atmospheric circulation (e.g., Gates et al., 1999; Eyring et al., 2016; Ackerley et al., 2018; Priestley et al., 2023).

Two ICON simulations used in this study are retrieved from the Climate Change Adaptation Digital Twin of the Destination Earth initiative ([https://destination-earth.eu](https://destination-earth.eu/)), which aims to develop operational, multi-decadal, storm-resolving simulations to support climate change impact assessments and adaptation planning at local to regional scales (Hoffmann et al., 2023; Sandu, 2024). These simulations follow the nextGEMS configuration and are marked in Table 1 with an asterisk (\*).

Regarding simulation periods and forcings, the historical simulations (ICON hist, IFS hist) cover 1990–2019 (for ICON, only data from 1993–2019 are available) and follow the CMIP6 historical forcing protocol. Future simulations (ICON fut, IFS fut) span 2020–2049 and use greenhouse gas concentrations—including ozone—prescribed by the CMIP6 SSP3-7.0 scenario (O'Neill et al., 2016).

The ability of the nextGEMS simulations to reproduce large-scale circulation patterns is evaluated against ERA5 reanalysis data (Hersbach et al., 2020) for the period 1990–2019. To contextualize the performance of storm-resolving models within the broader climate modeling landscape, we additionally analyze simulations from eight CMIP6 models (see Table 2), which cover the historical period 1985–2014 and were used in Dolores-Tesillos et al. (2025). For consistency across datasets, all relevant variables are remapped to a common  $1^{\circ} \times 1^{\circ}$  spatial resolution before blocking identification. Note that most of the multidecadal simulations cover 30 years, which is expected to be sufficient for capturing natural variability and identifying robust blocking signatures (Gao et al., 2025).

| Model          | $\Delta x_{\rm A}$ (km) | $\Delta x_{\mathrm{O}}$ (km) | Period   | Forcing            |
|----------------|-------------------------|------------------------------|----------|--------------------|
| IFS AMIP       | 9                       | 5                            | 30 years | Observed SST/SIC   |
| IFS hist       | 9                       | 5                            | 30 years | Historical (CMIP6) |
| IFS SSP3-7.0   | 9                       | 5                            | 30 years | SSP3-7.0           |
| ICON hist*     | 10                      | 5                            | 27 years | Historical (CMIP6) |
| ICON SSP3-7.0  | 10                      | 5                            | 30 years | SSP3-7.0           |
| ICON SSP3-7.0* | 5                       | 5                            | 20 years | SSP3-7.0           |

**Table 1.** Overview of storm-resolving simulations performed with the ICON and IFS-FESOM (IFS) models. The ICON simulations originated from the Destination Earth initiative are marked with \*.  $\Delta x_{\rm A}$  and  $\Delta x_{\rm O}$  denote the atmospheric and oceanic horizontal grid spacing, respectively. "Observed SST/SIC" refers to prescribed sea surface temperatures and sea ice concentrations based on observations.

| Model           | $\Delta x_{\rm A}$ (km) | Period   | Forcing    | Member ID |
|-----------------|-------------------------|----------|------------|-----------|
| MRI-ESM2-0      | 100                     | 30 years | Historical | rli1p1f1  |
| ACCESS-CM2      | 250                     | 30 years | Historical | rlilp1f1  |
| EC-Earth3       | 100                     | 30 years | Historical | rlilp1f1  |
| MPI-ESM1-2-HR** | 100                     | 30 years | Historical | rli1p1f1  |
| CESM2-WACCM*    | 100                     | 30 years | Historical | rli1p1f1  |
| MIROC6          | 250                     | 30 years | Historical | rli1p1f1  |
| MPI-ESM1-2-LR** | 250                     | 30 years | Historical | rli1p1f1  |
| CESM2*          | 100                     | 30 years | Historical | rllilplfl |

Table 2. List of CMIP6 models used in this study, including their horizontal grid spacing ( $\Delta x_A$ ), simulation period, forcing, and member ID. Models marked with \* or \*\* share components or dependencies, as discussed in Brunner et al. (2020). The member ID encodes simulation variants based on realization, initialization, physics, and forcing configuration.

#### 3 Methods

# 3.1 Blocking identification and tracking

#### 3.1.1 Anomaly-based index (ANOM).

The ANOM index is calculated following the approach of Woollings et al. (2018), consistent with earlier studies (e.g., Schwierz et al., 2004). It identifies blocking by tracking anomalies in the 500 hPa geopotential height field ( $Z_{500}$ ). The methodology consists of the following steps:

- A daily climatology is computed as the mean  $Z_{500}$  for each calendar day across each decadal baseline period (e.g., 1990–1999, 2000–2009, and 2010–2019), thereby removing interannual variability and long-term trends. A 31-day running mean is then applied to the daily  $Z_{500}$  data within each decade to smooth short-term fluctuations.

- Anomalies are obtained by subtracting the decadal climatology from the corresponding daily  $Z_{500}$  values.
  - To remove high-frequency variability, the anomaly fields are smoothed using a 2-day running mean. A blocking threshold is then defined as the 90th percentile of  $Z_{500}$  anomalies over the 50–80°N latitude band.
  - Candidate blocking events are identified when the anomaly exceeds this threshold. These events must also satisfy persistence and quasi-stationarity criteria, requiring a spatial overlap of at least 50% between consecutive days for a minimum duration of five days.

## 3.1.2 Absolute index (ABS).

Instantaneous blocks (IBs) are identified based on reversals in the meridional geopotential height gradient, following the method of Brunner and Steiner (2017) and consistent with previous work (e.g., Tibaldi and Molteni, 1990; Scherrer et al., 2006; Davini and D'Andrea, 2016; Rohrer et al., 2018). Three gradients are computed at each longitude  $\lambda$  and central latitude  $\phi$ :

$$\Delta Z_N(\lambda, \phi) = \frac{Z(\lambda, \phi + \Delta\phi) - Z(\lambda, \phi)}{\Delta\phi} \tag{1}$$

$$\Delta Z_S(\lambda, \phi) = \frac{Z(\lambda, \phi - \Delta\phi) - Z(\lambda, \phi)}{\Delta\phi}$$
(2)

$$\Delta Z_E(\lambda, \phi) = \frac{Z(\lambda, \phi - 2\Delta\phi) - Z(\lambda, \phi - \Delta\phi)}{\Delta\phi}$$
(3)

For the Northern Hemisphere, a grid point is classified as blocked if it satisfies the following conditions:

$$\Delta Z_N < -10$$
 m (°lat. $^{-1}$ ) 
$$\Delta Z_S < 0$$
 m (°lat. $^{-1}$ ) 
$$\Delta Z_E > 5$$
 m (°lat. $^{-1}$ )

Here,  $\lambda$  spans longitudes from 180°W to 179°E, and  $\phi$  latitudes from 75°S to 75°N. Gradients are calculated using  $\Delta \phi = 15^{\circ}$ . Following previous studies (e.g., Tibaldi and Molteni, 1990; Davini et al., 2012; Prodhomme et al., 2016; Davini and D'Andrea, 2020), no temporal or spatial filtering is applied to the instantaneous blocking field. This ensures a larger sample size and preserves the detailed spatial structure of blocking events. Notably, results remain qualitatively similar when filtering is applied, as discussed in Davini and D'Andrea (2020).

## 3.2 Blocking metrics

To characterize blocking events, we compute several metrics from the identified blocking fields. **Frequency** is defined as the number of blocked days per season or year at each grid point, providing spatial and temporal information on blocking occurrence. **Duration** refers to the total length of each blocking episode, while **size** denotes the spatial extent of the blocking pattern, calculated as the physical area (in km<sup>2</sup>) of all contiguous grid points meeting the blocking criteria, with variations in grid-box size by latitude accounted for. These metrics allow for a comprehensive comparison of blocking characteristics across models and scenarios. Analyses are presented for the main seasons: December to February (DJF) and June to August (JJA).

#### 160 3.3 Storm-tracks

To quantify synoptic-scale weather variability, we apply a Lanczos bandpass filter (2–6 days) to daily geopotential height at 500 hPa (Hoskins and Hodges, 2002; Greeves et al., 2007; Davini et al., 2017). The resulting bandpass-filtered fields are denoted by a prime ('). storm-tracks are identified based on the standard deviation of the filtered geopotential height ( $Z'_{500}$ ), which serves as a proxy for the intensity, frequency, and location of transient eddies.

Storm-track intensity is computed seasonally for boreal winter (DJF) and summer (JJA), providing insight into the spatial and temporal variability of synoptic activity across different models and time periods.

#### 4 Results

180

## 4.1 Northern Hemisphere winter blocking in multidecadal simulations

In this section, we compare blocking frequency, duration, and size in multidecadal storm-resolving simulations (IFS AMIP, IFS hist, and ICON hist) with the CMIP6 multi-model mean and the ERA5 reanalysis. Our focus is on Northern Hemisphere winter (DJF), using the ANOM index to highlight robust and spatially coherent biases. Additional results for the ABS index are provided in the Supplementary Material (Figs. S1 and S2). The Northern Hemisphere summer season (JJA) is discussed in the next section.

Figure 1 shows DJF blocking frequency biases relative to ERA5. IFS simulations reduce the underestimation of blocking in the North Atlantic found in CMIP6 (Fig. 1a,b) by about 10%. However, this improvement comes at the cost of an overestimation in other regions, including the Pacific and continental sectors. The RMSE values in Table 3 confirm that IFS AMIP and IFS hist outperform the CMIP6 ensemble in the North Atlantic. Note that comparing IFS AMIP with CMIP6 is not fully fair, since the AMIP run uses observed SSTs, whereas the CMIP6 results are taken from coupled simulations that include an ocean model.

In contrast, ICON exhibits a pronounced underestimation of blocking frequency and an eastward displacement of blocking maxima (Fig. 1d) over the North Atlantic. Its North Atlantic RMSE (1.89) is higher than that of both IFS hist (1.21) and the CMIP6 ensemble (1.42). Among the IFS simulations, the AMIP run performs best, suggesting that biases in the coupled version are partly driven by sea surface temperature (SST) errors. When observed SSTs are prescribed, the Atlantic blocking bias is substantially reduced (Fig. 1b)—an improvement even more evident when using the ABS index (Fig. S1a).

195

In the North Pacific, IFS hist tends to overestimate blocking frequency, while AMIP simulations again show reduced biases. ICON hist underestimates Pacific blocking but overestimates activity over California, consistent with IFS AMIP. To assess the effect of ensemble averaging, we also compute the mean bias of ICON and IFS hist (Fig. 1e). This combined field appears smoother and exhibits RMSE values close to the CMIP6 ensemble mean (Table 3), suggesting that part of the CMIP6 ensemble's skill may arise from compensating biases among models.

Blocking frequency biases can result from differences in event number, size, or duration. Figures 2a,d show the number of blocking events per year in each basin. In the North Atlantic, the IFS AMIP and CMIP6 means match ERA5 reasonably well, while ICON and IFS hist underestimate event counts. In the Pacific, IFS hist and ICON produce more divergent results: IFS hist underestimates the number of blocking events but overestimates frequency, implying longer-lived blocks, while ICON shows the opposite behavior.

Blocking duration metrics (Figs. 2b,e) indicate that IFS simulations tend to overestimate mean duration in the North Atlantic but capture the 95th percentile relatively well in IFS AMIP. ICON hist is closer to ERA5 in terms of mean duration and comparable to the CMIP6 ensemble. In the North Pacific, IFS hist generates fewer but longer blocks than ERA5, whereas IFS AMIP produces slightly shorter events. ICON yields more frequent but shorter-lived blocks. This supports the interpretation that Pacific frequency biases in IFS hist stem from persistent blocks, while ICON's biases are linked to smaller and shorter-lived events.

Blocking size is another key feature (Figs. 2c,f), as larger blocks typically lead to more persistent and widespread impacts (e.g., Nabizadeh et al., 2019). In the North Atlantic, the IFS simulations overestimate blocking size, which may contribute to frequency overestimation. ICON simulates smaller blocks, with a distribution that deviates more from ERA5 and CMIP6. In the Pacific, IFS hist tends to overestimate and ICON to underestimate size, whereas CMIP6 closely matches ERA5 for this metric.

## 205 In summary, we find that:

- IFS generally outperforms ICON, especially in the North Atlantic.
- IFS AMIP achieves the best agreement with ERA5, particularly in blocking frequency, highlighting the importance of accurate SST forcing.
- In the Pacific, IFS hist simulates fewer but longer-lasting blocks, while ICON produces more but shorter-lived blocks.
- ICON performs worse than CMIP6 for most metrics, especially with respect to blocking frequency and size.

These findings suggest that, while higher resolution alone does not guarantee improved blocking representation, storm-resolving models with realistic boundary conditions (as in AMIP) provide a clearer pathway toward better simulation of blocking. This result also emphasizes the need to better understand the interplay between SST biases, background flow, and moist processes in shaping blocking characteristics.

**Figure 1.** Blocking frequency biases against ERA5 during Northern Hemisphere winter (DJF), based on the ANOM index, for (a) IFS historical, (b) IFS atmosphere-only, (d) ICON historical, (e) the CMIP6 ensemble mean based on 8 models, and (g) the mean of IFS and ICON historical simulations. Black contours indicate ERA5 blocking frequency (4% intervals starting at 4%). Hatched areas highlight regions with relative differences exceeding 80%. Black dots indicate statistically significant differences (Z-test in panels (a), (b) and (c); model agreement ≥ 80% in panels (d) and (e)). Blue and red outlines indicate the North Atlantic and North Pacific basins, respectively.

**Table 3.** Root Mean Square Error (RMSE) of DJF blocking frequency in historical simulations relative to ERA5, quoted separately for the North Atlantic and North Pacific sectors.

| Simulation      | Atlantic RMSE | Pacific RMSE |
|-----------------|---------------|--------------|
| IFS hist        | 1.21          | 1.20         |
| IFS AMIP        | 1.06          | 1.04         |
| ICON hist       | 1.89          | 1.53         |
| IFS + ICON hist | 1.55          | 1.36         |
| CMIP6 ensemble  | 1.42          | 1.23         |

**Figure 2.** Number of blocking events and their properties in the (a,b,c) North Atlantic basin and (d,e,f) North Pacific basin during DJF. The extent of the basins is shown as dashed lines in Fig. 1, with blue representing the Atlantic and red the Pacific basin. Boxes represent the interquartile range (Q1-Q3), with the horizontal line indicating the median, whiskers extending from the 5th to the 95th percentile, and the red dot denoting the mean. Hatched boxes indicate statistically significant differences relative to ERA5 based on a Mann-Whitney U test (p 

220

## 5 4.1.1 The representation of the background flow

We first examine how biases in the large-scale zonal flow relate to winter blocking characteristics in the multidecadal storm-resolving models. As a diagnostic of the background flow, we analyze the seasonal-mean zonal wind at 500 hPa.

Figure 3 shows DJF mean 500 hPa zonal-wind biases relative to ERA5. In IFS hist, a positive bias of 2–3 m s<sup>-1</sup> is found over the eastern North Atlantic and Europe. In the North Pacific, the jet is displaced poleward, with positive wind biases along its northern flank and negative biases to the south. These features are consistent with the typical link between stronger or shifted zonal flow and reduced blocking frequency. For instance, in the North Atlantic region, the positive westerly bias coincides with negative blocking anomalies (cf. Fig. 1).

As discussed in the introduction, blocking occurrence is related to a decrease in waveguide strength in the zonal direction. We use  $\partial u/\partial x$  as a simple proxy. Stippling in Fig. 3 highlights regions of strong zonal-wind deceleration ( $\partial u/\partial x < -0.3 \times 10^{-5} \text{ s}^{-1}$ ) for each model, while ERA5 zonal-wind deceleration is shown as gray hatching. These areas—often associated with jet-exit zones—can indicate leaky waveguides that may facilitate wave breaking and block formation (Nakamura and Huang, 2018). In winter, the spatial alignment between jet-exit regions and blocking anomalies offers a useful sanity check on the realism of large-scale flow features in the models: blocking is suppressed in regions of strong westerlies but may be favored near the jet exit, where wave breaking and diffluence increase.

IFS AMIP shows a better jet representation (both strength and location) relative to IFS hist, particularly in the North Atlantic, where the RMSE drops from 1.92 m s<sup>-1</sup> (IFS hist) to 0.88 m s<sup>-1</sup> (IFS AMIP) (Table 6). In contrast, ICON substantially overestimates jet strength in both basins. This bias likely contributes to the underestimation of blocking frequency and the predominance of short-lived blocking events in ICON (Figs. 1, 2).

IFS storm-resolving simulations overcome a common limitation of CMIP6 models—namely, the equatorward bias in jet latitude (e.g., Dolores-Tesillos et al., 2025). IFS captures the latitudinal jet position better over the North Atlantic than CMIP6 (Fig. S3). However, when evaluated using RMSE, only IFS AMIP outperforms the CMIP6 ensemble mean in both basins. The historical coupled runs of ICON and IFS show larger overall wind biases (Table 6).

Figure 3. Mid-level (500 hPa) zonal-wind biases against ERA5 in Northern Hemisphere winter for (a) IFS historical, (b) IFS atmosphere-only, (c) ICON historical, and (d) the CMIP6 ensemble mean (8 models). ERA5 zonal wind is indicated by contours. Hatched areas indicate regions where the bias relative to ERA5 exceeds 80%. Regions of strong zonal-wind deceleration  $(\partial u/\partial x 

**Table 4.** Root Mean Square Error (RMSE) of 500 hPa zonal wind relative to ERA5 for different historical simulations, computed over the Northern Hemisphere Atlantic and Pacific basins (DJF).

| Simulation          | Atlantic RMSE (m s <sup>-1</sup> ) | Pacific RMSE (m s <sup>-1</sup> ) |
|---------------------|------------------------------------|-----------------------------------|
| IFS historical      | 1.92                               | 2.61                              |
| IFS AMIP            | 0.88                               | 0.99                              |
| ICON historical     | 3.50                               | 5.65                              |
| MRI-ESM2-0          | 3.00                               | 4.13                              |
| ACCESS-CM2          | 2.31                               | 2.39                              |
| EC-Earth3           | 1.23                               | 1.93                              |
| MPI-ESM1-2-HR       | 2.53                               | 2.93                              |
| CESM2-WACCM         | 1.36                               | 1.13                              |
| MIROC6              | 2.56                               | 3.47                              |
| MPI-ESM1-2-LR       | 2.43                               | 2.45                              |
| CESM2               | 1.63                               | 1.35                              |
| CMIP6 ensemble mean | 2.13                               | 2.47                              |

## 4.1.2 The representation of the ocean

240

250

255

We compare the coupled IFS hist simulations with their atmosphere-only counterparts (IFS AMIP) to examine how SST anomalies influence the location and frequency of winter blocking. Additionally, we use ICON and CMIP6 simulations as references to contextualize the role of ocean forcing across different model frameworks. Notably, the SST biases averaged over the eight CMIP6 models used here are similar to those of the full CMIP6 and CMIP5 ensembles (Zhang et al., 2023).

We structure this section by basin (North Atlantic and North Pacific), focusing on SST biases and their influence on baroclinicity, the jet stream, and blocking. Within each region, we compare simulations from IFS (coupled and AMIP) and ICON to highlight shared mechanisms and model-specific responses. Finally, we summarize cross-basin insights and contrasts with CMIP6 behavior.

In the North Atlantic, both IFS and ICON simulations exhibit pronounced SST biases during DJF, particularly in the midlatitudes (Fig. 4). Negative SST anomalies are found off Newfoundland and south of Greenland, enhancing the zonal SST gradient at midlatitudes. This strengthens lower-tropospheric baroclinicity and supports a stronger eddy-driven jet near 50–55°N. Simultaneously, warm anomalies north of the Gulf Stream tend to weaken the meridional SST gradient farther south, potentially reducing baroclinicity and limiting jet development in that region. The combined effect of these biases results in a poleward and eastward displacement of the eddy-driven jet, which favors more zonal flow and inhibits the development of persistent blocking patterns. These results are consistent with the mechanism proposed by Scaife et al. (2011) and further supported by Athanasiadis et al. (2022); Cheung et al. (2023). Despite exhibiting similar SST-bias patterns, CMIP6 models tend to simulate an equatorward-biased jet. This discrepancy suggests that model resolution and the representation of eddy–mean-flow interactions are critical for accurately capturing the circulation response to SST anomalies.

260

In the North Pacific, IFS hist simulations exhibit positive SST biases in the midlatitudes. These biases increase the ocean–atmosphere temperature contrast, likely enhancing surface latent-heat fluxes and contributing to increased lower-tropospheric baroclinicity through moist destabilization (Hermoso et al., 2024). The primary atmospheric response is not a uniform strengthening of the jet, but a poleward shift in its position (Fig. 3a), which coincides with a poleward shift in blocking frequency.

In contrast, ICON displays a more pronounced meridional SST gradient, with warm subtropical waters and cold polar waters. This sharper thermal contrast may enhance baroclinicity and intensifies the upper-level jet. Blocking events in ICON are generally shorter-lived, supporting the interpretation that strong zonal flow suppresses both the formation and maintenance of blocks.

The intensified and zonally extended jets in ICON (coupled with short-lived blocks and downstream-shifted blocking maxima) underscore the sensitivity of large-scale circulation to SST patterns in high-resolution coupled models. These results highlight the importance of accurately representing both oceanic boundary conditions and their coupling with atmospheric dynamics when simulating extratropical variability and blocking behavior.

**Figure 4.** SST biases against ERA5 in the Northern Hemisphere winter for (a) IFS historical, (b) ICON historical, and (d) the CMIP6 ensemble mean based on 8 models. The ERA5 SST is indicated by contours. Hatched areas indicate regions where the difference relative to ERA5 exceeds 80% in degree Celsius.

## **4.1.3** The representation of storm-tracks

Figure 5 shows the storm-track intensity biases during DJF. Compared with ERA5, the IFS hist simulation captures the main storm-track branches over both the North Pacific and North Atlantic. In the Atlantic, the storm-track is stronger and extends farther east than in ERA5. This eastward extension aligns with the eastward extension of the jet and the eastward shift of the

275

blocking maximum, and with the underestimation of blocking over the central North Atlantic. In the Pacific, the storm-track is shifted northward, consistent with the poleward displacement of the jets and the blocking frequency. These spatial associations suggest a coherent relationship among storm-track, jet, and blocking biases across regions.

The IFS AMIP simulation more accurately reproduces storm-track intensity in both the North Atlantic and the Mediterranean, along with a more realistic blocking distribution. ICON simulations exhibit an eastward-shifted and overly zonal storm-track in the North Atlantic—consistent with the stronger jet structure discussed earlier—and an eastward shift in blocking frequency. In the North Pacific, the storm-track is shifted northward, similar to the jets and the blocks.

The CMIP6 ensemble mean displays weaker and more equatorward storm-tracks than ERA5, especially in the Atlantic—a long-standing bias in coarser-resolution models (Zappa et al., 2013; Harvey et al., 2020; Priestley et al., 2023). This is consistent with their equatorward jet displacement and contributes to the underestimation of blocking frequency over the North Atlantic (e.g., Woollings et al., 2018).

Taken together, these results reaffirm the tight interplay between storm-tracks, jet structure, and atmospheric blocking. They underscore that realistic storm-track representation—particularly when supported by accurate ocean boundary forcing—is essential for capturing the dynamics of persistent weather regimes in the extratropics.

**Figure 5.** Storm-tracks biases against ERA5 in the Northern Hemisphere winter for (a) IFS historical, (b) IFS atmosphere-only, (c) ICON historical, and (d) the CMIP6 ensemble mean. The ERA5 storm-tracks amplitude is indicated by contours. Hatched areas indicate regions where the relative difference to ERA5 exceeds 80%.

## 4.2 Summer blocking in multidecadal simulations

We now turn to Northern Hemisphere summer (JJA) to assess the most robust and significant blocking biases across models. Note that summer biases (Fig. 6) are generally smaller than winter biases (Fig. 1).

300

305

In the North Atlantic, the IFS historical simulation overestimates blocking frequency, particularly south of Greenland, despite having an RMSE (0.62) similar to the CMIP6 ensemble mean (Table 5). This highlights a limitation of basin-averaged RMSE: it can mask strong localized anomalies. The pronounced overestimation of Greenland blocking in IFS hist (exceeding 80%) is largely absent in IFS AMIP, consistent with improvements also seen in the ABS index (Fig. S2b). The atmosphere-only configuration (IFS AMIP) achieves a lower RMSE (0.42) than IFS hist. ICON hist shows smaller frequency biases than IFS in the Atlantic region. ICON hist has a lower RMSE than the CMIP6 ensemble mean (0.33 vs. 0.60) and performs comparably well, ranking ahead of IFS AMIP in terms of basin-averaged error.

In the North Pacific, the IFS hist simulation underestimates blocking frequency, while IFS AMIP more closely matches ERA5. ICON hist exhibits a northward shift in blocking activity, leading to an overestimation at higher latitudes. The CMIP6 ensemble mean shows a different bias pattern, with a westward displacement of blocking maxima and an overall smaller magnitude than that seen in ICON.

We next examine the number of blocking events per year (Figs. 7a,d). In the North Atlantic, IFS simulations produce mean and median values close to ERA5. IFS hist shows the closest numerical agreement, while IFS AMIP slightly underestimates the number of blocks but remains the next-best performer. ICON simulations underestimate the number of events, consistent with their Atlantic frequency bias. In the North Pacific, IFS hist underestimates the number of blocking events, whereas IFS AMIP simulates slightly more. ICON simulations overestimate the block count, while the CMIP6 ensemble mean underestimates it.

Figures 7b,e present statistics for blocking duration. None of the model distributions differ significantly from ERA5, indicating that duration is generally well captured in JJA. In the North Atlantic, IFS simulations show a slight positive bias in the median duration, which may help explain their overestimation of blocking frequency (Fig. 6a). In the North Pacific, all storm-resolving models closely match ERA5 median duration, whereas CMIP6 exhibits longer durations, with IFS AMIP showing mildly overestimation values in the upper percentiles. ICON simulations reproduce ERA5 durations well, suggesting that their frequency overestimation likely stems from too many events rather than excessively long ones (Fig. 7d).

Figures 7c,f present statistics for blocking size. In the North Atlantic, IFS AMIP overestimates block size median, while IFS hist shows better overall agreement with ERA5. IFS AMIP produces the largest blocks on average, which, combined with its slightly longer-lived events, may partially offset its lower block count and contribute to a modest frequency overestimation. ICON simulations, as in winter, produce smaller blocks than ERA5. In the North Pacific, the IFS hist simulation reproduces ERA5 block size better than IFS AMIP. The AMIP simulation shows larger blocks, indicating that block size may be a limitation in an otherwise well-performing configuration. ICON simulations also slightly overestimate size in JJA, with values comparable to the CMIP6 ensemble mean.

The key findings are:

- IFS AMIP achieves the lowest RMSE for blocking frequency in the North Atlantic and shows balanced performance across duration and size, though it slightly overestimates block size and underestimates count.
- IFS hist best reproduces ERA5 median properties in the Atlantic but its Greenland blocking frequency is too high, likely because these blocks are too persistent.

- ICON shows realistic durations but underestimates frequency in the Atlantic and overestimates it in the Pacific: ICON also tends to simulate smaller blocks in both basins.
  - In the North Pacific, CMIP6 outperforms the storm-resolving models for upper duration percentiles, while IFS AMIP
    has the highest block count and overestimates size.
  - The CMIP6 ensemble underestimates blocking frequency and number of events but captures blocking duration well. In some cases, its performance is comparable to or better than the storm-resolving models.
- Taken together, these results show that storm-resolving simulations can improve aspects of summer blocking, especially when coupled with realistic SST forcing. However, their performance is not consistently better than the coarser-resolution models.

**Figure 6.** Blocking frequency biases against ERA5 during Northern Hemisphere summer (JJA), based on the ANOM index, for (a) IFS historical, (b) IFS atmosphere-only, (c) ICON historical, (d) the CMIP6 ensemble mean based on 8 models, and (e) the mean of IFS and ICON historical simulations. Black contours indicate ERA5 blocking frequency (4% intervals starting at 4%). Hatched areas highlight regions with relative differences exceeding 80%. Black dots indicate statistically significant differences (Z-test in panels (a), (b) and (c); model agreement ≥ 80% in panels (d) and (e)). Blue and red outlines indicate the North Atlantic and North Pacific basins, respectively.

**Table 5.** Root Mean Square Error (RMSE) of JJA blocking frequency in historical simulations relative to ERA5, separated for the North Atlantic and North Pacific sectors.

| Simulation      | Atlantic RMSE | Pacific RMSE |
|-----------------|---------------|--------------|
| IFS hist        | 0.62          | 0.58         |
| IFS AMIP        | 0.42          | 0.58         |
| ICON hist       | 0.33          | 0.51         |
| IFS + ICON hist | 0.48          | 0.55         |
| CMIP6 ensemble  | 0.60          | 0.68         |

Figure 7. Number of blocking events and their properties in the (a,b,c) North Atlantic basin and (d,e,f) North Pacific basin during JJA. The extent of the basins is shown as dashed lines in Fig. 1, with blue representing the Atlantic and red the Pacific basin. Boxes represent the interquartile range (Q1-Q3), with the horizontal line indicating the median, whiskers extending from the 5th to the 95th percentile, and the red dot denoting the mean. Hatched boxes indicate statistically significant differences relative to ERA5 based on a Mann-Whitney U test (p 

## 4.2.1 The representation of the background flow

In boreal summer (JJA), the climatological background flow differs markedly from winter, with generally weaker westerlies and a more zonally symmetric jet structure across the midlatitudes. Figure 8 shows the composite 500 hPa zonal wind and its bias against ERA5, while Table 6 presents RMSE values for the Atlantic and Pacific basins.

All storm-resolving simulations capture the seasonal weakening of the jet over the Atlantic. However, regional discrepancies remain. IFS hist shows a slightly negative wind bias southwest of Greenland (Fig. 8a), which coincides with a local overestimation of blocking frequency. In contrast, IFS AMIP and ICON simulate stronger-than-observed westerlies southeast of Greenland (Figs. 8b,d), consistent with reduced blocking in that region. These patterns broadly support the expected inverse relationship between jet strength and blocking occurrence.

Further downstream, all storm-resolving simulations exhibit a poleward shift of the jet over the eastern Atlantic, visible both in the spatial wind bias maps (Fig. 8) and the Atlantic zonal mean distribution (Fig. S4). Among the storm-resolving models, IFS AMIP performs best, with lower RMSE (0.99 m s<sup>-1</sup>) than IFS hist (2.39 m s<sup>-1</sup>) and ICON (2.06 m s<sup>-1</sup>), and with a more accurate representation of both jet structure and blocking activity.

In the North Pacific, IFS hist simulates broader and more intense jets over the western basin compared to ERA5 (Fig. 8a), with a clear equatorward displacement over the eastern basin (Fig. S4). The broader jet is also present in the CMIP6 ensemble mean, albeit less pronounced (Fig. 8d). IFS AMIP shows a more realistic jet latitude and intensity (RMSE =  $1.38 \text{ m s}^{-1}$ ), which is consistent with its improved blocking representation.

ICON hist, by contrast, features a strongly poleward-biased jet extending from Asia into the North Pacific, with particularly large wind biases over central Eurasia and the western Pacific (Fig. 8c). This jet configuration is associated with an underestimation of blocking over Eurasia and an overestimation at higher latitudes in the Pacific (cf. Fig. 6). Notably, in the Pacific sector, the overestimated blocking coincides with stronger-than-observed westerlies, in contrast to the more common inverse relationship between jet intensity and blocking occurrence. This highlights that the linkage between jets and blocking is not uniform: in some regions weaker jets favor more blocking, while in others stronger jets coincide with enhanced blocking frequency.

IFS AMIP performs best in reproducing observed jet structure and blocking activity in JJA, with the lowest RMSE across both basins and the most realistic meridional structure of blocking. In contrast, persistent jet latitude and intensity biases in ICON hist and IFS hist—particularly their poleward-displaced jets and excessive westerlies over Asia—contribute to their misrepresentation of blocking in both the Atlantic and Pacific sectors.

Figure 8. Mid-level (500 hPa) zonal wind biases against ERA5 in the Northern Hemisphere summer for (a) IFS historical, (b) IFS atmosphere-only, (c) ICON historical, and (d) the CMIP6 ensemble mean based on 8 models. The ERA5 zonal wind is indicated by contours. Hatched areas indicate regions where the bias relative to ERA5 exceeds 80%. Regions of strong zonal wind deceleration  $(\partial u/\partial x 

**Table 6.** Root Mean Square Error (RMSE) of 500 hPa zonal wind relative to ERA5 for different historical simulations, computed over the Northern Hemisphere Atlantic and Pacific basins (JJA).

| Simulation          | Atlantic RMSE (m s <sup>-1</sup> ) | Pacific RMSE (m s <sup>-1</sup> ) |
|---------------------|------------------------------------|-----------------------------------|
| IFS historical      | 2.39                               | 2.34                              |
| IFS AMIP            | 0.99                               | 1.38                              |
| ICON historical     | 2.06                               | 2.90                              |
| MRI-ESM2-0          | 1.63                               | 1.47                              |
| ACCESS-CM2          | 2.25                               | 2.22                              |
| EC-Earth3           | 1.55                               | 1.88                              |
| MPI-ESM1-2-HR       | 1.53                               | 1.51                              |
| CESM2-WACCM         | 1.54                               | 1.71                              |
| MIROC6              | 2.11                               | 1.57                              |
| MPI-ESM1-2-LR       | 1.92                               | 1.79                              |
| CESM2               | 1.40                               | 1.75                              |
| CMIP6 ensemble mean | 1.74                               | 1.73                              |

## 4.2.2 The representation of the ocean

365

In boreal summer (JJA), ocean–atmosphere coupling is generally weaker than in winter (e.g., Kushnir et al., 2002). This seasonal difference arises because the atmospheric circulation tends to be more barotropic and less sensitive to ocean surface anomalies. In addition, the shallower mixed layer depth reduces the ocean's thermal inertia and limits its ability to influence the atmosphere (e.g., Barsugli and Battisti, 1998). As a result, processes such as land-surface heating and convective activity often exert a stronger influence on the background flow than SST anomalies (e.g., Shaw and Voigt, 2015). Nonetheless, persistent and spatially coherent SST anomalies can still affect large-scale circulation and modulate blocking frequency during summer (e.g., Shaw and Voigt, 2015; Coumou et al., 2018; Osborne et al., 2020).

Figure 9 shows JJA SST biases against ERA5. In the North Atlantic, IFS hist exhibits negative SST anomalies co-located with the region of positive blocking frequency bias south of Greenland. One interpretation is that these cold anomalies suppress surface turbulent heat fluxes, leading to enhanced static stability and reduced baroclinicity. This may weaken the eddy-driven jet and favor the persistence of high-pressure systems (Häkkinen et al., 2011). In IFS AMIP, which uses observed SSTs, blocking biases over the North Atlantic are much smaller, suggesting that the SST biases in IFS hist likely contribute to the excessive blocking frequency.

In contrast, ICON hist displays warm SST anomalies across most of the North Atlantic (Fig. 9) with the exception of the central North Atlantic. Such positive anomalies would generally be expected to enhance surface fluxes and thereby intensify the eddy-driven jet, which in turn could suppress or shift blocking. In ICON, however, we do not find a clear jet strengthening; instead, the jet is slightly displaced poleward relative to ERA5, while blocking biases in this sector are weak and not statistically significant.

In the North Pacific, IFS hist simulates lower SSTs than ERA5, though the impact of this bias is less clear and discussed further in Section 5. By contrast, the ICON historical run exhibits widespread warming. The associated increase in baroclinicity appears linked to a stronger, poleward-shifted jet, accompanied by a northward displacement of blocking frequency. This suggests that the blocking response to SST anomalies is not straightforward but may depend on the interaction between SST patterns, the background flow, and eddy activity.

Taken together, these results suggest that even in summer, SST anomalies can influence blocking characteristics, particularly when biases are strong and spatially structured. Accurate representation of ocean surface conditions, either through coupling or prescribed SST forcing, remains important for reducing blocking biases in summer simulations.

**Figure 9.** SST biases against ERA5 in the Northern Hemisphere summer for (a) IFS historical, (b) ICON historical, and (d) the CMIP6 ensemble mean based on 8 models. The ERA5 SST is indicated by contours. Hatched areas indicate regions where the difference relative to ERA5 exceeds 80% in degree Celsius.

## **4.2.3** The representation of the storm-tracks

Figure 10 shows storm-track intensity biases during JJA. Compared to DJF, storm-tracks in summer are generally weaker and shifted poleward, consistent with reduced baroclinicity and a weaker jet stream (e.g., Zappa et al., 2013; Priestley et al., 2023; Harvey et al., 2020).

395

400

410

In the North Atlantic, the IFS coupled simulation (IFS hist) shows a storm-track that extends too far east compared to ERA5, with a localized intensity maximum south of Greenland—coinciding with the region of enhanced blocking frequency. IFS AMIP most closely resembles ERA5 in both storm-track location and intensity over the Atlantic and outperforms the CMIP6 ensemble. These small biases align with the improved blocking representation in IFS AMIP. ICON simulations feature a weaker storm-track in this region.

In the North Pacific, all models simulate a storm-track that is stronger at the jet entrance and weaker at the jet exit compared to ERA5, with ICON additionally indicating a slight poleward shift. In IFS and CMIP6 simulations, the weakened storm-track over the eastern North Pacific coincides with reduced blocking. The meridional displacement in ICON is likewise reflected in the blocking patterns, with ICON overestimating blocking frequency at higher latitudes. These results suggest that the wintertime relationship—where a weaker jet corresponds to fewer blocks—does not hold universally across all seasons and regions.

Overall, while summer storm-tracks are weaker and more meridionally spread compared to winter, their spatial structure still influences blocking occurrence. Among the storm-resolving models, IFS AMIP shows the best agreement with ERA5 storm-track patterns.

To further quantify the relationship between storm-track intensity and blocking frequency, we compute the spatial correlation of their biases relative to ERA5 across models (Figure 11) capturing the spatial co-variability between storm-track and blocking bias patterns. In both the North Atlantic and North Pacific, most CMIP6 models exhibit moderate positive correlations (ensemble mean: r=0.55 and r=0.59, respectively), suggesting a consistent spatial link between storm-track intensity and blocking frequency. The storm-resolving models show more diverse behavior. IFS hist shows weak-to-moderate correlations (Atlantic r=0.34, Pacific r=0.52), while ICON hist exhibits very weak correlation in the Atlantic (r=0.09) but moderate correlation in the Pacific (r=0.51). Interestingly, IFS AMIP shows near-zero correlation in the Atlantic (r=0.02), indicating that storm-track intensity alone does not fully explain blocking behavior in this configuration.

These results suggest that while a consistent spatial link between storm-track and blocking biases exists—especially in CMIP6 models—the storm-resolving models reveal more complex, regionally dependent dynamics likely influenced by variations in jet structure, eddy activity, and the background flow.

**Figure 10.** Storm-tracks difference to ERA5 in the Northern Hemisphere summer for (a) IFS historical, (b) IFS atmosphere-only, (c) ICON historical, and (d) the CMIP6 ensemble mean. The ERA5 storm-tracks amplitude is indicated by contours. Hatched areas indicate regions where the relative difference to ERA5 exceeds 80%.

**Figure 11.** Spatial correlation between storm-track and blocking biases in boreal summer (JJA) for selected simulations, including individual CMIP6 models and storm-resolving (SR) models. Correlations are computed across grid cells within the North Atlantic and North Pacific.

## 4.3 Climate change insights

420

After evaluating present-day blocking biases, we next examine projected changes under future climate conditions. Specifically, we analyze the IFS SSP3-7.0 simulations to explore how blocking frequency and characteristics may evolve in response to anthropogenic forcing, with attention to seasonal and regional differences. These results should be interpreted cautiously, given the substantial biases in the historical simulations—particularly regarding jet structure, SST patterns, and blocking persistence. The ICON scenario runs were also analyzed but are shown only in the Supplementary Material. In the ICON nextGEMS simulation, the absence of SST warming implies that blocking frequencies mainly reflect internal variability, although they still provide a useful illustration of how storm-resolving models simulate blocking. In the ICON DestinE simulation, the historical (10 km atmosphere / 5 km ocean—ice) and future (5 km across components) runs differ in resolution, making it difficult to

disentangle scenario signals from resolution effects. Nevertheless, we document these results in the Supplementary Material, as they still provide insight into ICON's representation of blocking (see Figs. S5 and S6).

Figure 12 presents projected changes in blocking frequency for (a) DJF and (b) JJA. In winter (DJF), IFS projects a reduction in high-latitude blocking frequency, particularly over northern Europe and eastern Asia, consistent with previous studies (Woollings et al., 2018). This decline is accompanied by increases of blocking frequency at lower latitudes, especially over the Pacific, suggesting an equatorward shift in North Pacific winter blocking. The order of magnitude of the changes is similar to the frequency biases in IFS. Note that the color scale in Figure 12 differs from that in Figure 1.

In summer (JJA), IFS simulates a decrease in blocking frequency across the mid-latitudes most prominently over the central North Atlantic and northern Europe. IFS simulates an increase in blocking frequency at higher latitudes, particularly around Greenland. The reduced blocking south of Greenland coincides with a warmer North Atlantic (see Fig. 13d), suggesting a possible thermodynamic link between increased surface temperatures and reduced blocking occurrence in this region, as discussed in Section 4.2.2. In general, the ICON SSP3-7.0 simulations reproduce the biases of the ICON historical run—for example, the eastward displacement of Atlantic blocking and the pronounced lack of North Atlantic blocking (Fig. S5).

Beyond frequency, we also assess changes in blocking characteristics such as duration and size (Figs. 12). Overall, these properties do not differ significantly from historical values across most regions. However, closer inspection reveals contrasting seasonal trends. In summer, IFS projects blocks that are slightly larger and longer-lived, especially over the Atlantic. In contrast, during winter, blocks are projected to be smaller and shorter-lived, particularly in the Atlantic.

Figure 12. IFS blocking response (SSP3-7.0 minus historical) during (a) winter and (b) summer. Black contours indicate IFS historical blocking frequency (2% intervals starting at 2%). Hatched areas highlight regions with relative differences exceeding 80%. Blue and red dashed outlines indicate the North Atlantic and North Pacific basins, respectively. Blocking properties in the North Atlantic (blue) and North Pacific (red) basins during (c–e, i–k) winter and (f–h, l–n) summer. Boxes represent the interquartile range (Q1–Q3), horizontal lines the median, whiskers the 5th–95th percentiles, and red dots the mean. Hatched boxes denote statistically significant differences relative to ERA5 (Mann–Whitney U test, p 

Projected changes in jet and storm-track structure under SSP3-7.0 are small compared to the model biases (Figs. 13). In general, the winter jet becomes slightly more zonal, while in summer it shifts poleward, consistent with previous studies (e.g. Harvey et al., 2020). In summer, over the northwestern Atlantic, IFS indicates a strengthening of the jet and more intense storm-tracks downstream, coinciding with regions where blocking frequency decreases. In winter, the largest response occurs in the Pacific, with an intensification of the subtropical jet and storm-tracks in areas where blocking frequency increases. However, the magnitude of these changes is small, making it difficult to disentangle the relative roles of the different processes.

Taken together, the SSP3-7.0 simulations suggest a reduction in Atlantic blocking in summer and an increase in North Pacific blocking in winter. While jet changes are modest, the consistent spatial shift in blocking points to a possible role for thermodynamic drivers, including Arctic amplification (altering meridional temperature gradients) and enhanced land–sea contrast (modifying regional heat fluxes and baroclinicity). These results underscore the need for process-based evaluation to disentangle the interplay between SST anomalies, jet dynamics, and blocking in a warming climate. The diversity in model responses—particularly in block duration and structure—highlights the importance of multi-model intercomparisons and high-resolution coupled simulations to build confidence in future blocking projections.

Figure 13. IFS projected changes in (a, b) background flow, (c,d) SSTs, and (e,f) storm-tracks under SSP3-7.0. Contours indicate IFS historical magnitude. Regions of strong zonal wind deceleration  $(\partial u/\partial x 

#### 5 Discussion

460

465

480

485

Our results indicate that increased horizontal resolution can improve several aspects of atmospheric blocking representation in climate models, particularly over the North Atlantic and Europe but large biases remain. These conclusions are based on 30-year simulations, an interval sufficient to capture key features of blocking variability (Gao et al., 2025). The storm-resolving IFS and ICON simulations reproduce several regional blocking characteristics more realistically than the CMIP6 ensemble mean. In particular, IFS AMIP captures block size and duration during boreal winter (DJF) with notable fidelity, suggesting that high resolution combined with realistic SST forcing enhances blocking persistence. These improvements are consistent with findings from idealized experiments (Schemm, 2023; De Luca et al., 2024) and high-resolution CMIP6 simulations (Schiemann et al., 2017, 2020; Gao et al., 2025). However, improvements are not uniform: ICON underestimates North Atlantic blocking frequency in both seasons, while some CMIP6 models match or outperform the high-resolution models in terms of blocking duration or spatial extent in specific regions.

Sea surface temperatures (SSTs) and ocean–atmosphere coupling exert a critical influence on midlatitude circulation and blocking behavior. At storm-resolving resolution, mesoscale SST gradients and frontal zones are better resolved, strengthening air–sea coupling (Wills et al., 2024; Vivant et al., 2025) and enhancing the sensitivity of the atmosphere to SST biases.

Among the IFS configurations, the atmosphere-only simulation (IFS AMIP), which uses observed SSTs, most closely reproduces ERA5 blocking characteristics—especially in terms of duration during DJF and frequency over the Greenland and North Pacific sectors. This highlights the importance of realistic SST boundary forcing. By contrast, IFS hist exhibits negative SST anomalies in the North Atlantic, particularly in summer, coinciding with regions of excessive blocking. One interpretation is that these cold anomalies suppress surface fluxes and stabilize the lower troposphere, thereby weakening baroclinicity and transient eddy activity and favoring the persistence of high-pressure systems (Häkkinen et al., 2011; Athanasiadis et al., 2022).

ICON simulations, on the other hand, display positive SST biases across both the North Atlantic and North Pacific. In winter, these warm anomalies coincide with intensified eddy-driven jets and reduced blocking, broadly consistent with the expectation that stronger baroclinicity suppresses blocking. In summer, however, the linkage is less straightforward: ICON shows a poleward-displaced rather than clearly intensified Atlantic jet, and the associated blocking reduction is weak and not always significant. This seasonal contrast suggests that while the jet–blocking relationship aligns with theoretical expectations in winter, it breaks down in summer, when weakened meridional SST gradients and altered jet latitude complicate the blocking response.

Yet SSTs do not act in isolation. Their impact is modulated by the background flow, transient eddies, and moist processes. For example, despite stronger SST gradients, ICON's overly zonal winter jet may inhibit ridge amplification, reducing blocking even where SSTs might otherwise support it. Similarly, in the North Pacific, SST front anomalies in ICON are colocated with suppressed storm-track activity and anomalous blocking at higher latitudes, pointing to nonlocal and nonlinear ocean–atmosphere interactions. Pacific SST gradients can even influence wave activity and blocking downstream over the Atlantic sector (Cheung et al., 2023; Hermoso et al., 2024).

500

505

510

515

Together, these results underscore the importance of accurate SST representation—both in magnitude and gradient—for simulating blocking. In summer, a consistent tendency emerges: colder North Atlantic SSTs (as in IFS hist) coincide with enhanced blocking south of Greenland, whereas warmer conditions (IFS AMIP and ICON) are associated with reduced blocking. Although the mechanisms differ across models, this contrast suggests that a warmer Atlantic background state generally favors fewer blocks. High-resolution models must therefore not only resolve mesoscale ocean features but also minimize SST biases to reliably capture blocking and its climate sensitivity.

The structure and variability of the eddy-driven jet and storm-tracks are closely linked to blocking behavior. In storm-resolving simulations, blocking frequency biases often coincide with jet latitude and intensity anomalies relative to ERA5. For example, in winter, the too zonal and poleward-shifted jets in ICON hist are associated with reduced blocking, while the more realistic jet configuration in IFS AMIP is linked to better agreement with observations.

Jet exit regions and zones of strong meridional jet curvature are particularly important for blocking development. In both IFS and ICON, deviations in jet position and strength relative to ERA5 correspond spatially to biases in blocking frequency, especially in the North Atlantic and Pacific sectors. These interactions are further shaped by storm-track variability: enhanced transient eddy activity can trigger blocking through wave breaking, whereas suppressed storm-tracks are often a consequence of the blocked flow, reflecting reduced eddy forcing when the background jet becomes unfavorable for synoptic development.

In summer, however, this coupling is less straightforward. Both IFS and ICON show storm-track suppression that is not accompanied by stronger blocking, indicating that eddy weakening alone does not systematically favor blocking. By contrast, coarse-resolution CMIP6 models tend to show a simpler relationship, with reduced storm-track activity occurring in regions where blocking also declines.

These results support previous findings that transient eddy activity is essential for the onset of blocking but does not uniquely determine its persistence (Hassanzadeh et al., 2014). The influence of storm-tracks on blocking is strongly conditioned by the broader circulation, including jet configuration, baroclinicity, and thermodynamic forcing (e.g., Zappa et al., 2014; Brayshaw et al., 2008). Blocking variability therefore arises from the coupled dynamics of jets, storm-tracks, and background flow features. Future studies should treat these components jointly, rather than in isolation, to better capture their mutual interactions and feedbacks under present and future climate conditions.

Despite the benefits of higher spatial resolution and more realistic SST forcing, biases in blocking representation remain in the IFS amip simulations. These remaining biases could be due to the representation of moist processes, including warm conveyor belts (WCBs) and diabatic heating (Schemm and Wernli, 2014; Dolores-Tesillos et al., 2025). Diabatic heating associated with WCBs can amplify ridges, reshape potential vorticity gradients, and precondition the flow for blocking onset (e.g., Pfahl et al., 2015). Inadequate simulation of these processes likely contributes to underestimations of blocking intensity and longevity.

IFS employs semi-parameterized convection schemes, whereas ICON disables deep convection parameterization altogether, making moist processes explicitly resolved. These contrasting treatments introduce different sensitivities to latent heating and may help explain the divergent blocking responses across models under historical and future forcing.

Further research should explicitly examine the role of moist processes in blocking dynamics within storm-resolving frameworks. Particular attention should be paid to vertical heating structures, cloud-radiative interactions, and the coupling between latent heat release and Rossby wave development. Strengthening this process-level understanding is essential to reduce persistent biases and improve the credibility of blocking projections under climate change.

In addition to evaluating present-day biases, our analysis provides insights into how blocking characteristics may evolve under future warming. The IFS SSP3-7.0 simulations suggest a latitudinal redistribution of blocking activity, with a decline in high-latitude winter blocking and reduced summer blocking over the North Atlantic midlatitudes.

These projected changes in blocking are accompanied by changes in jet structure and storm-tracks. In winter, IFS shows an intensification of the subtropical Pacific jet and storm-tracks that coincides with increased blocking frequency. In summer, over the northwestern Atlantic, the jet strengthens and storm-tracks intensify downstream in regions where blocking becomes less frequent. Thus, while an inverse blocking–storm-track relationship holds in some regions and seasons (e.g., the North Atlantic in summer), it is not universal; the winter Pacific provides a clear counterexample. Regarding blocking intensity and persistence, IFS projects slightly larger and longer-lived summer blocks under SSP3-7.0. The trend during winter is less clear, but a slight decrease in size and persistence is projected.

Taken together, these results indicate that future blocking behavior will likely be shaped by a combination of thermodynamic drivers (e.g., Arctic amplification, SST warming and gradients), large-scale jet shifts, and changing storm-track dynamics. Process-based evaluation and multi-model intercomparisons remain essential for improving confidence in these projections.

To build on these advances, future research should prioritize the following directions:

- Development and evaluation of coupled storm-resolving Earth system models, including multi-decadal simulations
  to assess blocking variability, climatological trends, and model stability across timescales.
- Improvement of moist process parameterizations particularly those governing convection and cloud-radiative feedbacks to more accurately capture their role in blocking onset, maintenance, and decay.
- Targeted diagnostics of moist dynamics, such as warm conveyor belt activity and vertical diabatic heating structures,
   to better understand their contributions to ridge amplification and flow reconfiguration.
  - Systematic analysis of storm-track-blocking coupling and SST-jet interactions, to clarify how transient eddies and
    ocean surface conditions jointly modulate blocking frequency and persistence under climate change.
  - Sensitivity experiments using perturbed SST fields, to isolate the relative contributions of oceanic versus atmospheric
    drivers in shaping jet structure and blocking behavior—especially in the context of high-resolution coupled models.
  - Enhanced representation of ocean-atmosphere coupling, with a focus on improving the exchange of heat, moisture, and momentum at the surface, as well as resolving SST gradients and mixed layer processes critical for blocking and jet variability.

These research avenues will be essential for leveraging the full potential of storm-resolving modeling and for addressing the persistent uncertainties that affect our understanding and simulation of midlatitude atmospheric circulation in a warming climate.

Storm-resolving modeling offers a promising path toward more realistic simulations of atmospheric blocking and other persistent weather extremes. However, our results underscore that increased spatial resolution alone is not sufficient. Meaningful progress will require a combined approach pairing high-resolution dynamics with realistic surface boundary conditions and targeted improvements in key physical processes, particularly moist dynamics and air—sea interactions. Advancing these elements in tandem will be essential for capturing blocking behavior more accurately and improving confidence in future projections under climate change.

#### 6 Conclusions

This study evaluates the representation of atmospheric blocking in storm-resolving climate simulations from the nextGEMS, EERIE and DestinE projects, comparing them to ERA5 reanalysis and a CMIP6 multi-model ensemble. We assess blocking frequency, duration, and spatial extent during boreal winter (DJF) and summer (JJA) across the Northern Hemisphere. The analysis includes both coupled and atmosphere-only simulations using the IFS and ICON models, enabling a systematic examination of resolution effects, ocean boundary conditions, and large-scale dynamics. Finally, we also examined the blocking response under SSP3-7.0 in IFS.

570 Our key findings are:

- Blocking biases remain in storm-resolving models: Relative to CMIP6, storm-resolving IFS and ICON simulations show both moderate improvements and deteriorations. Improvements are most apparent in the North Atlantic during winter, where the typical equatorward jet bias is reduced and blocking persistence is better captured (especially in IFS AMIP). However, in many other regions the storm-resolving simulations exhibit biases that are of comparable or even larger magnitude than those in CMIP6. Thus, higher resolution does not systematically translate into improved blocking climatologies.
- Resolution modulates jets and blocking through competing effects: Higher resolution helps sharpen mean-flow gradients and jet structures, which can locally improve blocking representation. However, differences between IFS and ICON and across seasons reveal that these benefits depend strongly on model physics and coupling strategy rather than grid spacing alone. Some CMIP6 models still match or outperform the storm-resolving configurations in individual metrics, underlining that resolution alone is not a guarantee of better blocking.
- Model differences are substantial: IFS simulations, particularly in the AMIP configuration, tend to agree more closely with ERA5 in terms of blocking frequency, size, and persistence. ICON, by contrast, often produces overly zonal jets and underestimates blocking in the Euro-Atlantic region. The comparison between IFS AMIP and coupled CMIP6 should be interpreted with caution, since no parallel AMIP analysis is done for CMIP6.

- Ocean boundary conditions strongly modulate blocking: IFS AMIP simulations highlight the importance of realistic SST forcing, with reduced blocking biases compared to the coupled version. In summer, colder SSTs south to Greenland are associated with a blocking increase. Across models, warm and cold SST biases affect blocking differently depending on their location relative to climatological gradients, which in turn influences jet strength and latitude. These impacts are neither spatially uniform nor seasonally consistent.
- Jet structure and storm-tracks shape blocking behavior: Blocking is closely linked to the configuration of the eddy-driven jet and storm-tracks. Strong, zonally aligned jets often coincide with reduced blocking, while jet curvature and jet-exit regions favor blocking occurrence. Although storm-track anomalies and blocking biases are not always colocated, positive correlations are found in some regions. Overall, their interaction is modulated by jet characteristics, baroclinicity, and seasonality.
- Climate change projections are similar to previous studies: IFS SSP3-7.0 projects a decline in high-latitude winter blocking and reduced summer blocking over the midlatitude North Atlantic, with some indication of a poleward displacement in summer. These changes are accompanied by modest jet and storm-track adjustments, though the relationship with blocking is not uniform. For example, in the winter Pacific, blocking and storm-track activity both intensify. Differences in projected block size, persistence, and spatial pattern underline the need for caution when interpreting these results. Overall, blocking appears sensitive to shifts in jet position, storm-track intensity, and SST forcing under warming, underscoring the importance of process-based evaluation and multi-model comparisons to build confidence in projections.

In summary, storm-resolving simulations represent a promising step toward a more process-based treatment of blocking. Yet, our findings show that resolution alone is not enough: improvements in some regions are offset by deteriorations elsewhere. Advancing blocking simulation will require not only higher resolution but also more realistic ocean boundary conditions, improved moist-process parameterizations, and coupled diagnostics of jets, storm-tracks, and blocking. Future work should therefore prioritize coupled storm-resolving Earth system models, targeted sensitivity experiments, and process-based evaluation across multiple models. These efforts are essential to reduce uncertainty, improve confidence in climate projections, and better assess the risks of persistent weather extremes in a warming climate.

Code availability. The code of blocking identification is available from https://github.com/steidani/ConTrack (Steinfeld, 2020).

*Data availability.* CMIP6, nextGEMS and EERIE data are available via easy.gems.dkrz.de. DesitinE runs can be retrieved from the DestinE Platform (https://platform.destine.eu, last access: 07 October 2025).

Author contributions. OM, SP and EDT designed the study. EDT performed the analysis, produced the figures, and drafted the manuscript.

All authors discussed the results and edited the manuscript.

*Competing interests.* At least one of the (co-)authors is a member of the editorial board of Weather and Climate Dynamics. The authors have no other competing interests to declare.

Acknowledgements. This work was supported by the EU Horizon 2020 Project nextGEMS, Grant Agreement Number 101003470. This work used resources of the Deutsches Klimarechenzentrum (DKRZ) granted by its Scientific Steering Committee (WLA) under project IDs bb1153 and bm1235. This work builds on analyses conducted as part of the Next Generation Earth Modelling Systems (nextGEMS) project. Portions of the manuscript are adapted from Deliverable D46 (Deliverable Release No. D9.1), titled "Report on the evaluation of atmospheric blocking and underlying mechanisms in the SR-ESMs." The author used OpenAI's language model (ChatGPT, https://chat.openai.com/, last access: 07 October 2025) to assist with grammar, phrasing, and consistency checks during manuscript preparation.

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
