# Peer review of "Storm-Resolving Models Advance Atmospheric Blocking Simulations and Climate Change Insights"

_EGUsphere, 2025_

## Referee Comment (RC2)

Review of WCD manuscript egusphere-2025-4969 "Storm-Resolving Models Advance Atmospheric Blocking Simulations and Climate Change" by Edgar Dolores-Tesillos et al. (2025)

**Summary**

The study illustrates how storm-resolving models represent European blocking using historical simulations of ICON and IFS-FESOM in comparison with eight CMIP6 models and ERA5. Despite regionally and seasonally-confined improvements, coupled storm-resolving model simulations reveal that typically known biases related to European blocking with respect to frequency, duration and spatial extent are not considerably reduced relative to CMIP models. Instead, an IFS AMIP Experiment forced with realistic SSTs indicates substantial improvement. Thus the study highlights the importance of realistically simulated SSTs for European blocking and emphasizes that a grid refinement towards the storm-resolving scale does not solve the European blocking bias problem. Another part of the study explains a negative trend of European blocking in specific regions, whereas the strong bias does not allow for reliable and accurate interpretations.

**Assessment**

The study provides interesting insights into the performance of storm-resolving models in simulating European blocking and associated trends. The study addresses different contributing factors, however, the study lacks a well-structured and consistent presentation of the results. Overall, there is a large number of major concerns that limits the study to be published in its current form. Given the scientific relevance of the study, I recommend a **major revision** of the manuscript by addressing the raised concerns before the manuscript can be considered to be accepted for publication in the journal of Weather & Climate Dynamics.

**General comments**

1.) The title can be misleading. The word 'Simulations' can be understood as climate model simulations, but this is not meant here. Either use the singular form ‚Simulation‘ or the word ‚representation' here. Also, the word 'advance' should not be used as the study concludes that the increased horizontal resolution in stom-resolving models does not lead to overall reduced blocking biases. Further, the phrasing 'Climate Change Insights' sounds very general and can eventually be tied to a specific result or removed due to reasons mentioned in the third comment.

2.) There is a general lack of motivation to several steps being performed in the study. For instance, the motivation about using storm-resolving models is too inadequate and requires further clarification of their advantages in the *Introduction* section. Further, several paragraphs need a re-structuring and enhanced consistency to guide the reader through the results more carefully.

3.) I consider the subsection 4.3 about the insights into European blocking trend too separate and too incomplete and would recommend removing this section from the study. Particularly, the fact that the European blocking biases largely remain in storm-resolving models complicate the assessment about their trends and therefore need a more careful consideration, which might be part of a follow-up study.

4.) The discussion section includes paragraphs, in which results are mentioned without being discussed in the context of related studies. Thus, the discussion appears too repetitious to me and needs to be more focused on how the results are related to the current research state.

5.) In general, the results of the included figures are not adequately and concisely presented. For some figures, certain subplots are not addressed in the results, even though they contain relevant features which would have been worth to be mentioned (see minor comments below). Further, there is an inconsistent and not convincing presentation as for Fig. 11. The figure is part of the subsection 4.2 explaining the blocking simulation during summertime, however I would expect such figure with a similar framework presented in the subsection 4.1 for wintertime as well.

6.) There are many incorrect or misplaced figure references. I highlight a couple of them in the detailed point-by-point responses below. Also, for the sake of a better readability, the authors should move figure references towards the end of a sentence.

Additional minor comments are provided in the line-by-line comments below.

**Minor comments**

**Line 7-17**: The paragraph appears somewhat unstructured due to jumps between AMIP and CMIP results. It would be more straightforward for the reader, if the paragraph is structured with the CMIP results first and followed by the AMIP results.

**Line 8**: Why do you choose 8 models from the CMIP6 ensemble? The authors should provide a reason why these models are chosen from the larger CMIP6 ensemble.

**Line 15-17**: How does this result relate to ERA5?

**Line 21**: The readability would be improved, if the paragraph is separated into two paragraphs here, i.e., separate the results related to the SSP3-7.0 forcing and the summary & implications from each other.

**Line 38-40 and following paragraphs**: The authors identify several factors contributing to the misrepresentation of blocking in climate models, namely (1) horizontal resolution, (2) storm-track biases, and (3) parameterisations of moist processes. However, the subsequent paragraphs introduce additional aspects (jet waveguide and SST representation) that are not included in the initial list. As also mentioned in major comment 2, I recommend revising this section for consistency by either following the original order of the listed processes or expanding the list to include all relevant factors. Providing a complete, numbered list upfront and then discussing each item in the same order would substantially improve clarity.

**Line 74 - 77:** The third and fourth research questions are very similar, and both refer to the final section on blocking representation under climate change. Since this section is intended as an addition with *first insights* rather than a main part of the paper, I

recommend raising only one research question for this section, if any, and combining the two existing questions into a single question.

**Line 89**: I recommend removing the phrasing 'the nextGEMS models', as it may lead to misunderstandings. Projects like nextGEMS do not own the models. They rather contribute to model development, conduct experiments, or use the models for research purposes.

**Line 92**: Why do you refer them to be storm-resolving Earth system models? The authors need to provide a justification. For instance, by stating the typical grid-spacing threshold below which a model is considered storm-resolving.

**Line 92-93**: This statement is too vague. The authors should clarify why the results are considered promising. Do the cited studies compare storm-resolving models with typical CMIP models and demonstrate an outperformance? The paper would benefit from a more detailed motivation for using storm-resolving climate models. This explanation could also be included in the introduction, which currently lacks a clear rationale for assessing storm-resolving models.

**Line 93**: '…for **the representation/simulation of** climate extremes…'

**Line 94**: 'A key distinction between **both** models'

**Line 116:** As mentioned above. Why do you use the subset of eight models from the CMIP6 ensemble.

**Line 139**: Is there a physical explanation or geographical direction associated with the three gradients that could be given here? I would assume they correspond to northern, southern, and equatorial gradients. Explicitly stating this would make the blocking index easier to understand.

**Line 159**: Why do the summer and winter seasons are the 'main seasons'. The authors may consider to replace this phrase by ‚winter and summer seasons'.

**Line 175**: For the comparison between IFS and CMIP, the authors have to refer to Fig. 1a,b,**d.**

**Line 179-180:** The authors should take care to describe the locations precisely. According to Fig. 1, I would rephrase the sentence to '…underestimation of blocking frequency over **the North Atlantic** and eastward displacement of blocking maxima **towards Eurasia.'**

**Line 180:** I would expect a reference to Fig. 1**c** rather than Fig. 1d.

**Line 186**: There is no compensation of biases over the North Atlantic and Eastern Europe (Fig. 1e). This should be stated here as well.

**Line 189:** Why does the blocking frequency biases result from the size and duration? To my understanding, the blocking frequency might only be influenced by size or duration if a blocked event is too small or too short-lived to be classified as a blocked event. Thus, the relationship emerges due to the constraints of the blocking indices rather than due to

a physical relationship/dependence. The authors should clarify the dependences of the blocking characteristics to each other as well as the influence of the methodology in order to avoid confusion.

**Line 194**: The sentence only describes the North Atlantic. Thereby, a reference to Fig. 2b is sufficient.

**Line 195**: '…but capture **mean and** the 95th percentile…'

**Line 197**: '…slightly shorter events (Fig. 2d,e).'

**Line 194-200**: This paragraph is intended to explain the models' blocking duration, as introduced at the outset. However, the subsequent sentences conflate duration, frequency, and size. I recommend guiding the reader systematically through these three diagnostics and then, in a separate paragraph, explaining the biases by drawing on the contributions of the blocking characteristics.

**Figure 1, caption**: Subplot labels are not correct. Please replace d), e) and g) by c), d) and e).

**Line 216**: The models are not multidecadal. The authors have to be more concise and may replace this part by '…in the multidecadal **simulations/experiments of both** storm-resolving models.'

**Line 237**: Do you refer to Table 4?

**Line 248**: 'Negative SST **bias…'**

**Line 270**: '…during **wintertime**.'

**Line 271**: ,'**Over** the **North** Atlantic,…'

**Line 291:** The areas on which the averages are performed, do not really fit to the terminology of 'North Atlantic' and 'North Pacific'. I would suggest to adjust and reduce the size of the areas to the ocean basins to avoid misunderstandings with the terminology.

**Line 294/295+ 303/304**: To my understanding, the conclusions about the performance of the ICON simulation does not agree in these two sentences. While the authors emphasize that ICON has a small blocking frequency bias (RMSE=0.33) (line 294/295), they describe an underestimation of the number of blocking events by referring back to the Atlantic frequency bias. Please clarify if the presentation of the results are correct. If so, I would recommend to revise the paragraphs with a more structured presentation of the results.

**Line 297**: ,…more closely…' If the authors refer with these findings to Fig. 6a,b, I would not conclude that IFS AMIP more closely matches ERA5. There are still significant differences with a similar magnitude compared to IFS hist over the North Pacific. The extent is only somewhat smaller in IFS AMIP. I would recommend a more concise presentation of the differences between the models with regard to blocking characteristics.

**Line: 301**: I would assume the authors only refer to IFS hist and not to both IFS simulations here. 'In the North Atlantic, **IFS hist produces** …'

**Line 316**: 'The AMIP simulation shows larger blocks **in general, i.e. the entire distribution is shifted towards an overestimation**,…' I would generally suggest to more concisely describe the distribution differences and highlight that not only the mean or median is different, but also the entire distribution is shifted relative to ERA5.

**Line 320-321:** ICON hist and not IFS AMIP has the lowest RMSE for blocking frequency in the North Atlantic according to Table 5. Also, what is meant by the term 'balanced performance'? Do we observe a compensation of biases between different blocking characteristics leading to an overall low RMSE? Please clarify the contribution of all three blocking characteristics.

**Line 325**: ICON does not simulate smaller blocks in the Pacific according to Fig. 7f.

**Line 326-327:** ICON-hist and not IFS AMIP has the highest block count over the North Pacific according to Fig. 7d**.**

**Line 332:** '…. than the coarse-resolution **CMIP** models.'

**Fig. 7d:** Where is the line for the mean of IFS amip?

**Figure 7, caption**: '…and their properties'. Providing the explicit list of properties/ characteristics with subplot references would be helpful here, e.g., count (a,d), duration (b,e), size (c,f)

**Line 335**: '…midlatitudes.' The authors need to provide a reference to a figure of the their study or cite a previous study.

**Line 339-340**: IFS hist shows a positive U bias here as well (Fig. 8a).

**Line 346:** '…more intense jets…' Do you refer to multiple jets here? Subtropical and polar jet? Please clarify.

**Line 347/348**: To my understanding, CMIP indicates a slight equatorward shift rather than a broader jet over Eurasia.

**Line 348/349**: The bias in CMIP is generally lower than in IFS AMIP, however, the RMSE for IFS amip is generally lower than for CMIP. Does this inconsistency emerges due to a stronger cancellation of biases with opposite signs in IFS amip and thus the limitation of the large size of areas, upon which the averaged RMSE values are calculated on?

**Line 353**: The overestimation of blocking largely occurs over North America and thus downstream of the poleward jet shift. Thus, it would not contradict the common inverse relationship between jet intensity and blocking occurrence.

**Line 369:** 'negative SST **bias**'

**Line 375**: 'warm SST **bias**'

**Line 376**: ‚Such positive **bias**'

**Line 380**: ‚the impact of this **cold** bias'

**Line 386**: ‚…when biases are strong…' - Why is the influence of SST anomalies on blocking only present when biases are strong? Bias differences in SST and blocking just indicate that a reduced SST bias might be related to a reduced blocking bias. Also what is meant by 'spatially structured'? I would recommend to remove the part after the comma.

**Figure 9, caption**: (c) instead of (d)

**Line 292**: Why is the acronym (IFS hist) introduced here again?

**Line 396**: ICON simulations feature a weaker storm track compared to IFS AMIP or ERA5?

**Line 403**: What is meant by 'meridionally spread'? This needs to be better expressed, for instance: ‚the jet is not exclusively tied to the mid-latitudes, and rather present on a wider latitude range' …do you mean something like that?

**Line 406 - 416 + Fig. 11**: There are a couple of shortcomings related to Fig. 11 and its description, as mentioned in major comment 5. Why does the figure illustrates only the relationship of blocking and storm tracks? Why is the relationship in this framework only shown for summer and not for winter in the previous section 4.1? The results are not really convincing as the relationship is relatively weak. Is this again a consequence of averaging over too large domain (smoothing and/or cancellation of signals)? I would recommend to revise the figure accordingly or consider removing it or move it to SI.

**Figure 9+10**: There is no hatching visible, eben though differences are quite large regionally. Please double-check if there is really no significance.

**Figure 11, caption**: What is meant by 'selected simulations'?

**Line 420/421+ line 443 +448**: The authors regularly point out that substantial biases must be taken into account when assessing blocking trends. As mentioned in the major comment 3, I would thus recommend to remove section 4.3 from the study.

**Line 431**: The suggested equatorward shift is not confirmed by a negative pattern directly northward of the positive pattern. The negative pattern appears over Eurasia (Fig. 12a).

**Line 435**: Fig. 13d clearly shows the typical North Atlantic warming hole. So the reduced blocking does not align with a North Atlantic warming pattern.

**Line 439-442**: The results on the number of events (count) should be presented here as well (Fig. 12c,f,i,l).

**Line 443-455:** Even though there is a small reference to it earlier in the text, this paragraphs lacks a more detailed discussion on the results related to the SST trend (Fig. 13c,d).

**Line 489-507 + Line 527-536**: These paragraphs are an extension of the results rather than a discussion of your results in the context of other studies. These paragraphs need to focus more on how the results relate to current research and should be shortened for clarity, as mentioned major comment 4.

**Line 515**: 'These remaining biases could be due to the **mis**representation of moist processes…'

**Line 541-553:** None of the raised bullet points include a reference to blocking under climate change. Instead, the most important future research branch motivated by the present study is the improvement of the representation of blocking in multi-decadal historical climate simulations to facilitate a more reliable estimate on how blocking will change regionally and seasonally under climate change conditions.

**Line 571 + 574/575**: This key finding indicate that storm-resolving models still have blocking biases and do not advance blocking simulation. As mentioned in the major comment 1, I would recommend revising the title of the study.

**Line 622**: Even though the authors mention the assistance of Large-Language Models (LLM) such as ChatGPT, I would suggest being careful to include longer sentence structures. Sentences in which an insertion is included separated by long dashes (em-dashes) is a typical output feature by ChatGPT.

---

## Author Comment (AC1)

**Response to Reviewers:**
**Storm-Resolving Models Advance Atmospheric Blocking Simulations and Climate Change Insights**

We thank the reviewers for the time and effort to evaluate this manuscript and for their constructive and insightful comments. We sincerely appreciate the suggestions provided, which have helped to improve the clarity and robustness of the study. All reviewer comments have been carefully considered and addressed in the revised manuscript. The text below (shown in blue) documents the specific changes made in response to each comment. Unless otherwise stated, figure and line numbers refer to the original submitted manuscript.

**Reviewer 1**

Climate modelling is increasingly moving toward higher spatial resolution, and recent initiatives such as NextGEMS and Climate DT are at the forefront in this development. These projects use kilometre-scale global simulations based on the ICON and IFS modelling systems, producing among the most advanced high-resolution climate model datasets available to the research community at the moment.

As these simulations from Climate DT are becoming more and more available, it is likely that they will play an important role in future climate change assessments due to their ability to better resolve and represent atmospheric key processes such as convection and cyclone structures. Therefore, evaluating the systematic biases of these models is essential. This manuscript contributes to that effort by assessing blocking characteristics in ICON and IFS models. Thus, I believe that this manuscript provides a timely and valuable addition to the literature, and it will be useful for future studies, for example those assessing changes in windstorm frequency or intensity in these high-resolution models.

The authors analyze blocking characteristics and storm tracks from ICON and IFS historical simulations, as well as IFS atmosphere-only (AMIP) simulations, and compare them against ERA5 reanalysis for both northern hemisphere winter and summer seasons. They also compare the results with the CMIP6 ensemble mean. Overall, the study finds that the higher resolution in ICON and IFS models does not necessarily translate to reduced biases in blocking and storm track features. The IFS AMIP simulation, which uses observed sea surface temperatures (SSTs), exhibits the smallest biases, highlighting the importance of realistic SSTs in coupled climate model simulations.

I think the paper was well written, although some sections such as Discussion felt a bit too lengthy. I was also a bit puzzled about the differences between NextGEMS and Climate DT projects and model simulations. As I am by no means an expert on blocking dynamics, I provided only some minor comments related to the presentation and readability of the paper. I hope that the other reviewers can comment more on the methodological choices. Overall, I am happy to recommend publication of this paper after my relatively minor comments have been addressed.

We thank the reviewer for the overall positive and encouraging assessment of our manuscript. We fully share the reviewer's view that kilometre-scale global climate models will play an important role in future climate risk and impact assessments, and we appreciate the recognition of the relevance and timeliness of this work. We acknowledge that the distinction between the NextGEMS and Climate DT projects may not have been

sufficiently clear in the original manuscript, particularly as the two initiatives share modelling systems and, in some cases, simulations. As noted by the reviewer, this can indeed be confusing. In the revised manuscript, we have clarified the relationship between the two projects, explicitly describing Climate DT as a follow-up initiative building upon the developments of NextGEMS, while highlighting their respective objectives and simulation setups. These clarifications have been incorporated in the revised text, with additional details provided in response to the minor comments below.

Minor comments:

1. It wasn't entirely clear for me what the differences are between NextGEMS and Climate DT simulations. Is the IFS FESOM SSP3-7.0 that you analysed the same simulation that will be used also in Climate DT? (it wasn't marked with star in Table 1). In the Supplementary material (Fig. S5-S6) you analysed the ICON NextGEMS and ICON DestinE SSP3-7.0. Is the only difference between them the atmospheric resolution? For example, one clarifying paragraph about the differences between NextGEMS and Climate DT / DestinE could work here.

   We thank the reviewer for pointing out that the distinction between the NextGEMS and Climate DT (DestinE) simulations was not sufficiently clear in the original manuscript. We agree that this clarification is important. The IFS–FESOM SSP3-7.0 simulation analysed in this study is the same simulation that is also made available through the Climate DT (Destination Earth) data portal [https://platform.destine.eu]. This simulation was originally produced within the NextGEMS project and was subsequently incorporated into the Climate DT dataset. We have clarified this explicitly in the revised manuscript and in Table 1. In the Supplementary Material (Figs. S5–S6), we analyse two ICON SSP3-7.0 simulations: one produced within the NextGEMS framework and one associated with Climate DT. According to the available documentation, the primary difference between these two ICON simulations is the atmospheric horizontal resolution, while the overall model configuration remains otherwise comparable. We have clarified this point in the revised text. More generally, we now include a dedicated clarifying paragraph in the Data section describing the relationship between NextGEMS and Climate DT / DestinE. In this paragraph, we explain that Climate DT builds upon developments and simulations from NextGEMS, while also including additional or updated simulations, depending on the modelling system and experimental design. This addition is intended to improve transparency and avoid confusion of readers. New proposed paragraph:
   "The simulations analysed in this study originate from two closely related European initiatives: nextGEMS and the Climate Change Adaptation Digital Twin (Climate DT) of Destination Earth. The Climate DT builds directly on developments from the nextGEMS project, including shared model configurations and, in some cases, identical simulations. In particular, the IFS–FESOM SSP3-7.0 simulation analysed here was originally produced within the nextGEMS framework and is also distributed through the Climate DT data portal. For ICON, both nextGEMS and Climate DT provide SSP3-7.0 simulations with comparable model configurations; however, the Climate DT ICON simulation analysed here employs a higher atmospheric horizontal resolution than its nextGEMS counterpart."

2. As far as I know, the Climate DT project also uses the IFS-NEMO model, which has a different ocean component (NEMO model) than in IFS-FESOM. Why didn't you analyse that model in this paper? It might be worth mentioning why it has been left out.

   We thank the reviewer for this pertinent question. We agree that the IFS–NEMO configuration used within the Climate DT (Destination Earth) initiative represents an important modelling system and is of high relevance. The primary reason for not including the IFS–NEMO simulations in this study is the limited temporal coverage available at the time the analysis was conducted. When this work was initiated, the IFS–NEMO Climate DT simulations did not yet provide a sufficiently long, continuous period of output to allow for a robust assessment of atmospheric blocking statistics, which typically requires multidecadal records to reliably sample internal variability. In addition, the present study focuses on ensuring a consistent comparison across models and experiments with respect to simulation length and forcing protocols. Given these constraints, we chose to limit the analysis to simulations with approximately 30 years of data, which we consider a minimum requirement for identifying statistically

robust blocking characteristics. We have now explicitly mentioned the existence of the IFS–NEMO Climate DT simulations in the revised Data section and clarified why they were not included in the present analysis. This point is also discussed in the new proposed paragraph describing the similarities and differences between the NextGEMS and Climate DT initiatives. New proposed paragraph:
"In addition to the IFS–FESOM configuration, the Climate DT initiative also includes simulations based on the IFS–NEMO coupled model. These simulations are not analysed in the present study because, at the time of analysis, they did not yet provide a sufficiently long and continuous temporal coverage to support a robust assessment of atmospheric blocking statistics. To improve transparency, simulations distributed through the Climate DT are explicitly marked in Table 1."

3. Throughout the results section, you talk about the North Atlantic and North Pacific basins but it seems that these domains cover the land as well, so they effectively mean the whole hemispheres. For example, Fig. 2a "Blocks in ATL": does that include the whole hemisphere from -90° to 90°E, and not just the North Atlantic ocean basin?

We thank the reviewer for this important clarification. This comment is similar to the minor comment 3 from Reviewer 1. Here, I describe again our adjustments. The reviewer is correct that the North Atlantic and North Pacific domains used in our analysis extend over both oceanic and adjacent land regions and therefore encompass large longitudinal sectors of the Northern Hemisphere, rather than being limited strictly to the ocean basins. This domain definition was chosen to ensure capturing blocking regimes such as Ural blocking, which is especially prominent during boreal summer. In response, we have revised the domain definitions and reduced their spatial extent to better align with the ocean basins, thereby avoiding ambiguity in the terminology. The updated North Atlantic and North Pacific domains are now more consistent with common basin-based definitions and no longer extend excessively over adjacent continental region. We now show blocking characteristics for more standard Euro-Atlantic and North Pacific domains (e.g., Schiemann et al., 2020). In general, the results remain similar, however, we will adapt the text where necessary. New proposed paragraph:
"Blocking statistics are evaluated over two broad longitudinal sectors of the Northern Hemisphere, referred to here as the North Atlantic (ATL) and North Pacific (PAC) regions. These domains are defined following Schiemann et al. (2020) and encompass both oceanic and adjacent continental areas, rather than being restricted strictly to the ocean basins. This choice allows for a consistent comparison with previous blocking climatologies and ensures that continental blocking regimes, such as Ural blocking, are adequately represented, particularly during boreal summer. The exact longitudinal and latitudinal bounds of each domain are for ATL 50–90 N, -90–90 E and PAC 40–90 N, 120–240 E ."

4. The discussion section is relatively lengthy. It could help readability to divide it into smaller subsections, and maybe condense the text if you find that possible.

We thank the reviewer for this helpful suggestion. In response, we have restructured the Discussion section into several smaller, clearly defined subsections to improve readability and flow. The section is now organized into clearly defined thematic subsections, namely: (i) Blocking Representation in Storm-Resolving Models, (ii) Role of SST Biases and Air–Sea Coupling, (iii) Jet, Storm-Track, and Blocking Interactions, (iv) Influence of Moist Processes, (v) Implications for Future Blocking Changes and (vi) Final Remarks. In addition, we have condensed the text where possible to reduce redundancy while preserving the key scientific arguments and interpretations.

5. Title. When I finished reading the paper, I was honestly a bit disappointed about the models' skill in reproducing the blocking and storm track features, given the enormous effort put in these digital twin activities. The biases in many variables, especially in SSTs, were very pronounced in IFS and ICON hist runs. Overall, I got the feeling that the new high-resolution models do not appear to outperform the CMIP6 models. In this light, for me, the title "Storm-Resolving Models Advance Atmospheric Blocking Simulations" seems somewhat optimistic.

We thank the reviewer for this thoughtful comment. We agree that the original title may convey an overly optimistic impression regarding the performance of storm-resolving models in reproducing

blocking and storm-track characteristics. As highlighted by our results, increased resolution alone does not systematically eliminate biases, particularly in coupled simulations affected by ocean-atmosphere feedback errors, and storm-resolving models do not uniformly outperform CMIP6 models across all metrics. To better reflect the balanced and critical nature of our findings, we have revised the title to: "Atmospheric Blocking Representation in Storm-Resolving Climate Models under Historical and Future Forcing".

Line-by-line comments:

L90: What does "energetically consistent climate" mean?

We thank the reviewer for this comment, this concept was developed in the nextGEMS project. Hans et al., 2025 defined an energetically consistent climate as the state of a model consistent with the conservation of mass and energy with a top-of-the-atmosphere (TOA) energy balance close to 0 and no near surface temperature drift. This was one key achievement of the nextGEMS project. The control runs initially had strong drift towards cold or warm climates. By modifying cloud properties, the energetically consistent climate was achieved. We will included a short description in main text.

Table 1. Maybe add one column for the years in the periods?

We will add this column to Table 1.

L163: Storm-tracks

We have capitalized the letter S.

Fig. 2 caption: "the extent of the basins is shown as dashed lines." I did not see any dashed lines in Fig. 1. See also minor comment # 2 about the basins. Also, the dashed line in Fig. 2 which seems to be the ERA5 median, is missing from the caption.

We thank you the reviewer for noting this issue. We refer to the solid line in figure 1. We have corrected the caption. Furthermore, we have included the dashed line meaning in the caption, that is exactly what you mention, the ERA5 median.

L231: Table 4? Same also in L237.

We apologize for the typo, you are right the correct reference is Table 4. We have modified accordingly.

Fig. 3: Is the colorbar value -10 correct? The scale seems otherwise to go with 1 m/s interval. Same for Fig. 8.

Yes, the values of the colorbar in Figs. 3/8 are: -10, -5, -4, -3, -2, -1, 1, 2, 3, 4, 5, 10. But, we agree that this may lead to a misinterpretation, thus we have modified the color bar of figure 3/8 to:
-8,-7,-6, -5, -4, -3, -2, -1, 1, 2, 3, 4, 5, 6,7, 8

Fig. 4: The panels do not have b) but the caption has. Same in Fig. 9.

We apologize for the discrepancies between the figure and the caption, we have modified the caption as follows:
Figure 4. SST biases against ERA5 in the Northern Hemisphere winter for (a) IFS historical, (c) ICON historical, and (d) the CMIP6 ensemble mean based on 8 models. The ERA5 SST is indicated by contours. Hatched areas indicate regions where the difference relative to ERA5 exceeds 80% in degree Celsius.
Figure 9. SST biases against ERA5 in the Northern Hemisphere summer for (a) IFS historical, (c) ICON historical, and (d) the CMIP6 ensemble mean based on 8 models. The ERA5 SST is indicated by contours. Hatched areas indicate regions where the difference relative to ERA5 exceeds 80% in degree Celsius.

L276: Maybe add that the northward shift of storm tracks in the Pacific seems to occur in western Pacific

only.

I believe you refer to line 279. Thank you for pointing this out. We have modified the line 279 to: In the western North Pacific, the storm-track is shifted northward, similar to the jets and the blocks.

L304: What is the Atlantic frequency bias in ICON? I cannot see it from Fig. 6c.

There is some light blue (underestimation) to the North of Scandinavia, over Northern Russia and eastern Europe. But you are right, this signal is not too evident; we will rephrase the sentence.

L309: whereas CMIP6 exhibits longer durations. For me, it seems that the duration in CMIP6 in the Pacific (Fig. 7e) is the same as in other models.

We thank the reviewer for mentioning this. Indeed, the CMIP6 presents roughly the same duration compared to the other models. We have modified line 309 accordingly.

L311: ICON frequency overestimation is true only for high latitudes, but these boxplots (in Fig. 7) show the values for the whole domain, right? So can you actually use that argument as there is underestimation over the lower latitudes in the Pacific (Fig. 6c).

Yes, you are right, however, the larger signal in general is an overestimation, thus we would expect this to be reflected in the statistics. To make it consistent, we will remove this argument.

L338: The slightly negative wind bias southwest of Greenland seems to be a very minor feature, and the increased blocking in my opinion is more over southeast of Greenland (Fig. 6a) where the bias in winds is positive (Fig. 8a). But is positive wind bias consistent with increased blocking in the region then? Overall I think the wind biases near Greenland are very consistent between all three models.

We thank the reviewer for this careful assessment. We agree that the slightly negative wind bias southwest of Greenland is a relatively minor feature and should not be overemphasized. We have therefore rephrased the corresponding text to avoid overstating its significance. In the revised manuscript, we clarify that wind biases near Greenland are generally similar across the three storm-resolving configurations. We also note that jet variability alone may not fully explain regional blocking differences, and that other processes, such as moist dynamics, may contribute. New proposed sentence:
"All storm-resolving simulations capture the seasonal weakening of the North Atlantic jet. However, regional discrepancies remain. Over the jet entrance region, the jet position in the IFS historical simulation is shifted equatorwards, whereas the jet in ICON hist and IFS AMIP is located slightly more poleward. Southeast of Greenland, wind biases are broadly similar across the three models and are characterized by stronger-than-observed westerlies. In this region, a clear inverse relationship between jet strength and blocking occurrence is not evident (see again Fig. 6). This suggests that zonal wind biases alone are insufficient to explain regional differences in blocking frequency, and that additional processes, including moist dynamics, are likely to play a contributing role."

L340: Should be maybe Figs. 8b,c

We agree that the right panels are b and c. We have modified to "Figs. 8b,c"

L340: Reduced blocking? I don't see reduced blocking southeast of Greenland in Fig. 6 in these models.

We thank the reviewer for pointing this out and apologize for the confusion. We agree that the wording was misleading. In this context, we intended to convey that ICON hist and IFS AMIP exhibit a smaller blocking overestimation compared to IFS hist, rather than an actual reduction in blocking frequency relative to ERA5. To avoid ambiguity, we have removed this sentence from the revised manuscript.

L347: The broader jet is present also in the CMIP6 ensemble. Aren't the CMIP6 jet biases in the Pacific somewhat opposite to that in IFS hist? Fig. 8a vs. 8d?

We thank the reviewer for this insightful comment. We agree that a broader Pacific jet is also evident in the CMIP6 ensemble mean. In the revised manuscript, we clarify that while both IFS hist and CMIP6 exhibit a broadened jet, the detailed structure of the wind biases differs between them. In particular, IFS hist shows relatively stronger wind anomalies on the equatorward flank of the jet, whereas the CMIP6 ensemble mean exhibits comparatively stronger anomalies on the poleward side. These differences do not indicate an opposite bias in jet width, but rather reflect variations in the latitudinal distribution of wind anomalies that are likely influenced by ensemble averaging across structurally diverse CMIP6 models. We have revised the text accordingly to better distinguish between these features and to avoid ambiguity in the comparison

L369: I think negative SST anomalies are basically everywhere in the North Atlantic in IFS hist so they are not just co-located with positive blocking biases. The interpretation starting from L370 (low SSTs -¿ favoring high pressure systems) seems to hold only in the high-latitude North Atlantic. The relationship is pretty much the opposite in the Pacific and absent in ICON so I'm not sure how robust the interpretation is.

We thank the reviewer for this thoughtful comment. We agree that negative SST anomalies in IFS hist extend over much of the North Atlantic and are not confined solely to regions of enhanced blocking frequency. We also agree that the interpretation linking cold SST anomalies to increased blocking is primarily applicable to the high-latitude North Atlantic and does not hold in the same way for the North Pacific, nor is it evident in ICON. In the original manuscript, the causal chain linking SST anomalies, reduced baroclinicity, and enhanced blocking was overstated. In the revised version, we therefore restrict the Results section to a factual description of the co-occurrence of cold SST biases and enhanced Greenland blocking in IFS hist, without implying a direct causal mechanism. The potential role of SST-driven processes is now discussed more cautiously in the Discussion section as a possible contributing factor rather than a robust explanation.

L393: Maybe you mean here bias maximum?

We thank the reviewer for this clarification. Yes, we refer here to the maximum in the blocking frequency bias. We have revised the text accordingly to avoid ambiguity.

Fig. 11 title: Do you mean spatial correlation of blocking biases vs. storm-track biases? The first "biases" was missing which is why I first thought you meant the absolute blocking and not its bias.

We thank the reviewer for pointing this out and apologize for the misleading wording. Yes, Fig. 11 shows the spatial correlation between storm-track biases and blocking frequency biases. We have revised both the figure title and caption to explicitly include "biases" and avoid any ambiguity between absolute fields and their biases.

Fig. 11. Is the correlation calculated for ocean-only?

We thank the reviewer for this question. The spatial correlations shown in Fig. 11 (now Fig. S8 in the Supplementary Material) are not restricted to ocean-only grid points. They are computed over the same North Atlantic and North Pacific domains used for the blocking analysis, which include both oceanic and adjacent land regions. In the revised manuscript, these domains have been updated to more standard regional definitions to avoid ambiguity, and their exact latitudinal and longitudinal bounds are now explicitly stated in the figure caption. The correlations are therefore calculated consistently over these revised domains.

L435: The reduced blocking in Fig. 12b (JJA) seems to be co-located with the North Atlantic warming hole (Fig. 13d). Thus, I do not really agree with your reasoning about the warmer North Atlantic in this context.

We thank the reviewer for this careful observation. We agree that the reduced blocking shown in Fig. 12b (JJA) is co-located with the North Atlantic warming hole evident in Fig. 13d, and that this complicates the interpretation of a simple relationship between warmer SSTs and reduced blocking in this region. In the original text, our interpretation overstated the robustness of a thermodynamic link. In the revised manuscript, we therefore limit the Results section to a factual description of the spatial patterns, without

attributing causality. Any discussion of potential thermodynamic influences is now treated more cautiously and deferred to the Discussion section.

Fig. 13. The colorbar scale in e-f is not symmetric around zero. I would put that symmetric to have balanced view of the biases.

We thank the reviewer for this helpful suggestion. We have revised Fig. 13e–f to use a symmetric color scale, providing a more balanced and consistent visualization of the biases.

Figs. S5 and S6. I don't understand these figures. Why does the caption say biases? I thought the b) and c) panels represent the response for climate change (i.e. SSP3–7.0 2020-2049 minus historical)? i.e. similar to Fig. 12a-b but for ICON model. If that's the case, shouldn't the comparison be against the model's historical simulation and not ERA5, given the biases?

We thank the reviewer for highlighting this ambiguity. We agree that the presentation of Figs. S5–S6 was not sufficiently clear. Although panels (b) and (c) show differences for SSP3–7.0 simulations, these figures are intentionally presented as biases relative to ERA5 rather than as climate change responses relative to the model's historical simulation. This choice was made because a direct comparison between the ICON SSP3–7.0 simulations and ICON hist is not straightforward. First, the ICON SSP3–7.0 simulation from Climate DT employs a finer atmospheric horizontal resolution than ICON hist, making it difficult to disentangle climate change signals from differences in model configuration. Second, the nextGEMS ICON SSP3–7.0 simulation exhibits a pronounced cold drift during the initial years, while ICON hist shows a warm SST bias relative to ERA5. As a result, differences between SSP3–7.0 and historical ICON simulations would largely reflect model drift and internal variability rather than a robust climate change signal. For these reasons, we chose to evaluate the SSP3–7.0 simulations relative to ERA5 and to present these results in the Supplementary Material only, with the intention of providing qualitative context on model behavior rather than a quantitative assessment of climate change impacts. We have clarified this rationale in the revised manuscript and updated the captions of Figs. S5–S6 accordingly.

**Reviewer 2**

**Summary**

The study illustrates how storm-resolving models represent European blocking using historical simulations of ICON and IFS-FESOM in comparison with eight CMIP6 models and ERA5. Despite regionally and seasonally-confined improvements, coupled storm-resolving model simulations reveal that typically known biases related to European blocking with respect to frequency, duration and spatial extent are not considerably reduced relative to CMIP models. Instead, an IFS AMIP Experiment forced with realistic SSTs indicates substantial improvement. Thus the study highlights the importance of realistically simulated SSTs for European blocking and emphasizes that a grid refinement towards the storm-resolving scale does not solve the European blocking bias problem. Another part of the study explains a negative trend of European blocking in specific regions, whereas the strong bias does not allow for reliable and accurate interpretations.

We thank the reviewer for the careful reading of our manuscript and for the clear and balanced summary of our main findings. We agree with the reviewer's assessment that persistent biases in European blocking remain in the coupled storm-resolving simulations, and that increased spatial resolution alone does not resolve these deficiencies. We also concur that interpretations of future changes in blocking must be treated with caution given the magnitude of present-day biases. As emphasized in the manuscript, our climate change analysis is therefore intentionally limited and framed conservatively. Nevertheless, we find that some projected signals are qualitatively consistent with those reported in CMIP-based studies. We believe that documenting these similarities and differences is valuable for the community, particularly in the context of ongoing efforts to assess the strengths and limitations of storm-resolving climate models. In the following, we address the reviewer's specific comments point by point.

**Assessment**

The study provides interesting insights into the performance of storm-resolving models in simulating European blocking and associated trends. The study addresses different contributing factors, however, the study lacks a well-structured and consistent presentation of the results. Overall, there is a large number of major concerns that limits the study to be published in its current form. Given the scientific relevance of the study, I recommend a major revision of the manuscript by addressing the raised concerns before the manuscript can be considered to be accepted for publication in the journal of Weather & Climate Dynamics.

We thank the reviewer for the assessment of our manuscript and for recognizing the scientific relevance of the study. We agree that a clear, well-structured, and consistent presentation of the results is essential. In response to the reviewer's comments, we have substantially revised the manuscript to improve its structure, clarity, and coherence, particularly in the presentation and interpretation of the results. All concerns raised by the reviewer have been carefully considered and addressed in the revised version. Detailed explanations of the changes made in response to each comment are provided below.

**General comments**

1.) The title can be misleading. The word "Simulations" can be understood as climate model simulations, but this is not meant here. Either use the singular form "Simulation" or the word "representation" here. Also, the word "advance" should not be used as the study concludes that the increased horizontal resolution in stom-resolving models does not lead to overall reduced blocking biases. Further, the phrasing "Climate Change Insights" sounds very general and can eventually be tied to a specific result or removed due to reasons mentioned in the third comment.

We thank the reviewer for this important comment and agree that the original title could be misleading. In particular, we acknowledge that the use of the plural form "Simulations" and the word "Advance" may imply an improvement in blocking representation that is not supported by our results. We also agree that the phrase "Climate Change Insights" is overly general and may raise expectations beyond the specific scope of the analysis. In response to this and to similar feedback from Reviewer 1, we have revised the title to

more accurately reflect the content and conclusions of the study. The revised title reads:
"Atmospheric Blocking Representation in Storm-Resolving Climate Models under Historical and Future Forcing."
This formulation emphasizes evaluation rather than implied improvement, clarifies the focus on representation, and more precisely reflects the scope of the historical and future analyses presented in the manuscript.

2.) There is a general lack of motivation to several steps being performed in the study. For instance, the motivation about using storm-resolving models is too inadequate and requires further clarification of their advantages in the Introduction section. Further, several paragraphs need a re-structuring and enhanced consistency to guide the reader through the results more carefully.

We thank the reviewer for this constructive comment and agree that the motivation for using storm-resolving models required clearer articulation. In the revised manuscript, we have substantially strengthened the Introduction by expanding the discussion of the potential advantages of storm-resolving models for representing large-scale circulation, particularly through their improved representation of mesoscale processes, air–sea coupling, and moist dynamics. We now reference additional recent studies that explicitly discuss how these processes may influence blocking and related circulation biases. In addition, we have revised and restructured several sections of the manuscript to improve consistency and better guide the reader through the results. This includes clearer transitions between sections, a more logical progression of the analyses, and improved signposting of key findings. These changes are intended to enhance readability and ensure that the motivation and interpretation of each analysis step are clearly communicated.

3.) I consider the subsection 4.3 about the insights into European blocking trend too separate and too incomplete and would recommend removing this section from the study. Particularly, the fact that the European blocking biases largely remain in storm-resolving models complicate the assessment about their trends and therefore need a more careful consideration, which might be part of a follow-up study.

We thank the reviewer for this important comment and agree that the assessment of European blocking trends is complicated by the presence of substantial present-day biases in both storm-resolving and conventional climate models. We also agree that a comprehensive and fully robust analysis of blocking trends would require a more dedicated treatment and is beyond the scope of the present study. Nevertheless, we believe that it is still valuable to provide a cautious, clearly qualified first assessment of how blocking characteristics evolve in the storm-resolving simulations analysed here, particularly where these signals are consistent with those reported in previous CMIP-based studies. As the reviewer notes, traditional climate models also exhibit persistent blocking biases, yet they continue to form the basis of our current understanding of future circulation changes. In this context, documenting similarities and differences between storm-resolving simulations and CMIP projections provides useful information for the community. In response to the reviewer's concern, we have substantially revised Section 4.3 to reduce its scope and remove any causal or overconfident interpretations. The revised section now focuses on a descriptive comparison of projected changes, explicitly highlights the limitations imposed by present-day biases, and frames the results as exploratory rather than definitive. We also emphasize that a more detailed and process-based assessment of blocking trends in storm-resolving models is a natural subject for future work.

4.) The discussion section includes paragraphs, in which results are mentioned without being discussed in the context of related studies. Thus, the discussion appears too repetitious to me and needs to be more focused on how the results are related to the current research state.

We thank the reviewer for raising this important concern. We agree that parts of the original Discussion section were overly descriptive and insufficiently connected to the existing literature, which reduced its focus and interpretative depth. In response, we have substantially revised the Discussion to reduce repetition of results and to more explicitly relate our findings to previous studies. The revised version places greater emphasis on interpreting the results in the context of current understanding of atmospheric blocking, storm-track dynamics, air–sea coupling, and moist processes. In addition, as also suggested by Reviewer 1, we have restructured the Discussion into clearly defined subsections and condensed the text where possible to improve clarity, focus, and readability.

5.) In general, the results of the included figures are not adequately and concisely presented. For some figures, certain subplots are not addressed in the results, even though they contain relevant features which would have been worth to be mentioned (see minor comments below). Further, there is an inconsistent and not convincing presentation as for Fig. 11. The figure is part of the subsection 4.2 explaining the blocking simulation during summertime, however I would expect such figure with a similar framework presented in the subsection 4.1 for wintertime as well.

We thank the reviewer for this careful assessment and agree that the presentation of the results and figures required improvement. In the revised manuscript, we have systematically reviewed all figures to ensure that they are explicitly addressed in the Results section with relevant features clearly described and interpreted. Several result paragraphs have been expanded or reorganized accordingly, following the reviewer's minor comments. We also agree that the presentation of Fig. 11 was inconsistent. As noted by the reviewer, this figure was originally shown only for summer because the corresponding wintertime signal was weak and less spatially coherent. However, we acknowledge that this choice was not sufficiently justified and reduced the overall consistency of the Results section. To address this, we have moved Fig. 11 to the Supplementary Material, where we now also include the corresponding wintertime analysis for completeness. The main text has been revised accordingly to avoid overemphasizing this diagnostic and to maintain a consistent framework between the winter and summer analyses.

6.) There are many incorrect or misplaced figure references. I highlight a couple of them in the detailed point-by-point responses below. Also, for the sake of a better readability, the authors should move figure references towards the end of a sentence. Additional minor comments are provided in the line-by-line comments below.

We thank the reviewer for pointing out these issues. In the revised manuscript, we have carefully checked and corrected all figure references, including those highlighted in the detailed line-by-line comments. In addition, we have revised the text to place figure references consistently at the end of sentences where appropriate, in order to improve readability.

**Minor comments**

**Line 7-17**: The paragraph appears somewhat unstructured due to jumps between AMIP and CMIP results. It would be more straightforward for the reader, if the paragraph is structured with the CMIP results first and followed by the AMIP results.

We thank the reviewer for this helpful suggestion and agree that the original paragraph mixed CMIP6 and AMIP results in a way that reduced clarity. In the revised manuscript, we have restructured the paragraph to first summarize the CMIP6 results, followed by the storm-resolving coupled simulations, and finally the atmosphere-only (IFS AMIP) results. This reordering provides a clearer and more logical progression for the reader and improves the overall consistency of the Results section

**Line 8**: Why do you choose 8 models from the CMIP6 ensemble? The authors should provide a reason why these models are chosen from the larger CMIP6 ensemble.

We thank the reviewer for raising this point and agree that the selection of CMIP6 models should be clearly justified. The set of eight CMIP6 models analysed in this study was chosen to ensure consistency with our previous work Dolores-Tesillos et al. (2025), in which the role of dry and moist processes in Euro-Atlantic blocking biases was examined in detail. The model selection was guided by two main criteria. First, we focused on models that exhibit relatively good overall performance in the CMIP6 ensemble, based on the evaluation presented by Palmer et al. (2022). Second, the selected models provide the necessary three-dimensional atmospheric variables at multiple pressure levels required for diagnosing moist processes, including the identification of warm conveyor belts using the ELIAS 2.0 framework (Quinting and Grams, 2022). To improve transparency, we have now explicitly described these selection criteria in the Data section of the revised manuscript. We chose not to include this level of detail in the abstract to maintain conciseness.

**Line 15-17**: How does this result relate to ERA5?

We thank the reviewer for pointing out this inconsistency. We agree that the quoted sentence was ambiguous and did not clearly relate the diagnosed relationship to ERA5, which could mislead the reader. The intent was to describe differences in the spatial correlation of biases between blocking frequency and storm-track activity across model classes (as illustrated in Fig. 11 of the original manuscript), rather than an absolute relationship relative to ERA5. As this sentence was out of context and potentially confusing, we have removed it from the revised manuscript and adjusted the surrounding text accordingly.

**Line 21**: The readability would be improved, if the paragraph is separated into two paragraphs here, i.e., separate the results related to the SSP3-7.0 forcing and the summary & implications from each other.

We thank the reviewer for this helpful suggestion and agree that the readability of this section can be improved by separating the discussion of SSP3–7.0 results from the broader summary and implications. In the revised manuscript, we have split this paragraph into two parts: the first focuses on projected blocking changes under SSP3–7.0 forcing, while the second summarizes the overall implications for storm-resolving model performance. This restructuring improves clarity and better distinguishes results from interpretation.

**Line 38-40 and following paragraphs**: The authors identify several factors contributing to the misrepresentation of blocking in climate models, namely (1) horizontal resolution, (2) storm-track biases, and (3) parameterisations of moist processes. However, the subsequent paragraphs introduce additional aspects (jet waveguide and SST representation) that are not included in the initial list. As also mentioned in major comment 2, I recommend revising this section for consistency by either following the original order of the listed processes or expanding the list to include all relevant factors. Providing a complete, numbered list upfront and then discussing each item in the same order would substantially improve clarity.

We thank the reviewer for this helpful and well-founded comment and agree that the original presentation lacked structural consistency. In the revised manuscript, we have reorganized this part of the Introduction by providing a complete, numbered list of the main factors contributing to blocking biases upfront. The revised list includes: (1) horizontal resolution; (2) storm-track (transient eddy) activity; (3) the large-scale jet waveguide (mean-flow); (4) sea surface temperature (SST) biases and air–sea coupling; and (5) moist diabatic processes. The subsequent paragraphs have been revised to follow this order consistently, with each factor discussed in turn. This restructuring improves clarity, avoids introducing additional mechanisms without prior context, and better guides the reader through the physical processes motivating the present study.

**Line 74-77**: The third and fourth research questions are very similar, and both refer to the final section on blocking representation under climate change. Since this section is intended as an addition with first insights rather than a main part of the paper, I recommend raising only one research question for this section, if any, and combining the two existing questions into a single question.

We thank the reviewer for this helpful suggestion and agree that the third and fourth research questions overlap and both refer to the exploratory analysis of blocking under future forcing. In the revised manuscript, we have combined these two questions into a single, more focused research question. This change better reflects the limited and contextual nature of the SSP3–7.0 analysis and avoids overemphasizing this aspect of the study.

**Line 89**: I recommend removing the phrasing 'the nextGEMS models', as it may lead to misunderstandings. Projects like nextGEMS do not own the models. They rather contribute to model development, conduct experiments, or use the models for research purposes.

We thank the reviewer for pointing this out and agree that the original phrasing could be misleading. The nextGEMS project does not own the ICON or IFS-FESOM models, but rather contributes to their development and applies them within a coordinated experimental framework. In the revised manuscript, we have rephrased this sentence to avoid implying model ownership and to more accurately reflect the role of the nextGEMS project.

**Line 92**: Why do you refer them to be storm-resolving Earth system models? The authors need to provide a justification. For instance, by stating the typical grid-spacing threshold below which a model is considered storm-resolving.

We thank the reviewer for this important clarification request and agree that the terminology "storm-resolving Earth system models" requires explicit justification. In the revised manuscript, we now clarify both components of this term. We refer to ICON and IFS-FESOM as Earth system models because they are fully coupled systems including atmosphere, ocean, sea ice, and land components, and are designed to support additional Earth system processes such as aerosols and the carbon cycle. The term storm-resolving is used to indicate that the atmospheric horizontal grid spacing (5–10 km) lies within the range commonly considered sufficient to explicitly resolve the mesoscale structure of extratropical cyclones and associated storm dynamics. Previous studies have identified grid spacings of approximately 10 km or finer as a practical threshold for storm-resolving global models (Hohenegger et al., 2023), although some definitions adopt more restrictive thresholds (e.g., ∼4 km) for fully convection-resolving simulations (Prein et al., 2015). We have incorporated this clarification into the Data section of the revised manuscript to avoid ambiguity and to clearly state the basis for our terminology.

**Line 92-93**: This statement is too vague. The authors should clarify why the results are considered promising. Do the cited studies compare storm-resolving models with typical CMIP models and demonstrate an outperformance? The paper would benefit from a more detailed motivation for using storm-resolving climate models. This explanation could also be included in the introduction, which currently lacks a clear rationale for assessing storm-resolving models.

We thank the reviewer for this important comment and agree that the original statement was too vague. In the revised manuscript, we clarify why storm-resolving simulations are considered promising, while avoiding any implication of a systematic or universal outperformance relative to CMIP-class models. Specifically, we now emphasize that the potential value of storm-resolving models lies in their improved representation of physical processes that are known to influence large-scale circulation, rather than in guaranteed reductions of blocking biases. Recent studies have shown that kilometre-scale global models improve the simulation of mesoscale precipitation structures, extreme rainfall, and orographic processes, and benefit from a much more realistic representation of topography. This improved orographic realism is expected to affect planetary-scale Rossby waves and the background flow, which are central to blocking dynamics. Some studies explicitly compare storm-resolving simulations with CMIP6 models and demonstrate added value for weather extremes and mesoscale dynamics (e.g., Wille et al., 2025), although not specifically for blocking statistics. Following the reviewer's suggestion, we have expanded the Introduction to provide a clearer motivation for assessing storm-resolving climate models. New proposed sentence:
"Initial evaluations of kilometre-scale global simulations indicate promising improvements in the representation of several processes relevant to large-scale circulation, including mesoscale precipitation structures, extreme rainfall, and orographic forcing (e.g., Wille et al., 2025; Brunner et al., 2025; Poujol et al., 2025). For instance, Wille et al. (2025) demonstrated that storm-resolving models can outperform conventional CMIP-class models in simulating precipitation extremes. The much finer representation of topography and mesoscale dynamics may also improve the simulation of planetary-scale waves and the background flow. However, whether these advances contribute to a better representation of atmospheric blocking remains an open question, motivating the present analysis."

**Line 93**: '...for the representation/simulation of climate extremes...'

We thank the reviewer for pointing this out. We have revised the sentence to explicitly include the noun and now refer to "for the representation of " to improve clarity and precision.

**Line 94**: 'A key distinction between both models'

We thank the reviewer for pointing this out. We have revised the text accordingly and now use the phrasing "a key distinction between both models" for clarity.'

**Line 116**: As mentioned above. Why do you use the subset of eight models from the CMIP6 ensemble.

We thank the reviewer for reiterating this point. As discussed in response to the earlier comment (Line 8), the subset of eight CMIP6 models was selected to ensure consistency with our previous work Dolores-Tesillos et al. (2025) and based on documented model performance and data availability. Specifically, the selection was guided by (i) overall CMIP6 model performance as assessed by Palmer et al. (2022), and (ii) the availability of three-dimensional atmospheric variables at multiple pressure levels required for diagnosing some processes. To avoid redundancy, we have incorporated this explanation explicitly in the Data section of the revised manuscript and refer the reader there for details.

**Line 139**: Is there a physical explanation or geographical direction associated with the three gradients that could be given here? I would assume they correspond to northern, southern, and equatorial gradients. Explicitly stating this would make the blocking index easier to understand.

We thank the reviewer for this helpful suggestion and agree that explicitly stating the geographical and physical interpretation of the three gradients improves the clarity of the blocking index definition. In the revised manuscript, we now clarify that the gradients correspond to the meridional geopotential height gradients to the north, south, and equatorward side of the central latitude, following the standard interpretation of the Tibaldi–Molteni–type blocking indices (Tibaldi and Molteni, 1990). This additional explanation has been included directly in the text preceding the equations.

**Line 159**: Why do the summer and winter seasons are the "main seasons". The authors may consider to replace this phrase by "winter and summer seasons".

We thank the reviewer for this suggestion and agree that the phrasing "winter and summer seasons" is clearer and more appropriate. We have revised the text accordingly.

**Line 175**: For the comparison between IFS and CMIP, the authors have to refer to Fig. 1a,b,d.

We thank the reviewer for pointing out this incorrect reference. We have corrected the text to refer to "Fig. 1a,d".

**Line 179-180**: The authors should take care to describe the locations precisely. According to Fig. 1, I would rephrase the sentence to '...underestimation of blocking frequency over the North Atlantic and eastward displacement of blocking maxima towards Eurasia.'

We thank the reviewer for this helpful suggestion and agree that the original wording did not sufficiently specify the spatial characteristics of the bias. We have revised the sentence to more precisely describe the underestimation of blocking frequency over the North Atlantic and the eastward displacement of blocking maxima toward Eurasia, in line with Fig. 1.

**Line 180**: I would expect a reference to Fig. 1c rather than Fig. 1d.

We thank the reviewer for pointing out this incorrect reference. We have corrected the text to refer to "Fig. 1c".

**Line 186**: There is no compensation of biases over the North Atlantic and Eastern Europe (Fig. 1e). This should be stated here as well.

We thank the reviewer for this important clarification. We agree that the ensemble-mean bias does not indicate a compensation of biases over the North Atlantic and Eastern Europe. In these regions, both ICON and IFS exhibit biases of the same sign, with an underestimation of blocking frequency over the North Atlantic and an overestimation over Eastern Europe. We have revised the manuscript accordingly to explicitly state this regional behavior and to clarify that the apparent smoothing of the ensemble-mean field and the associated reduction in RMSE primarily reflect spatial averaging rather than true bias cancellation. New proposed paragraph:

To assess the effect of ensemble averaging, we also compute the mean bias of ICON and IFS hist (Fig. 1e). The resulting field appears spatially smoother and exhibits RMSE values comparable to the CMIP6 ensemble mean (Table 3). However, ICON and IFS display biases of the same sign over key regions (both underestimate blocking over the North Atlantic and overestimate it over Eastern Europe), indicating that the apparent smoothing does not reflect a compensation of biases in these regions but instead results from spatial averaging of similar model errors.

**Line 189**: Why does the blocking frequency biases result from the size and duration? To my understanding, the blocking frequency might only be influenced by size or duration if a blocked event is too small or too short-lived to be classified as a blocked event. Thus, the relationship emerges due to the constraints of the blocking indices rather than due to a physical relationship/dependence. The authors should clarify the dependences of the blocking characteristics to each other as well as the influence of the methodology in order to avoid confusion.

We thank the reviewer for this important clarification and agree that the relationship between blocking frequency, duration, and size requires a more careful explanation. As the reviewer correctly points out, blocking frequency is not an independent physical property, but rather an accumulated diagnostic that depends on how individual blocking events are identified and counted by the blocking index. In particular, frequency biases can arise not only from differences in the number of blocking events, but also from methodological effects related to event duration and spatial extent. Long-lived or spatially extensive blocking events contribute more blocked days and grid points, thereby increasing blocking frequency even if the total number of events is unchanged. Conversely, short-lived or spatially small events may fail to meet the persistence or size thresholds of the blocking index, leading to reduced frequency. We have revised the text to explicitly distinguish between physical differences in blocking behavior and dependencies introduced by the blocking identification methodology, in order to avoid confusion.

**Line 194**: The sentence only describes the North Atlantic. Thereby, a reference to Fig. 2b is sufficient.

We thank the reviewer for pointing this out. We have corrected the figure reference accordingly and now refer to "Fig. 2b".

**Line 195**: '. . . but capture mean and the 95th percentile. . . '

We thank the reviewer for this suggestion. We have revised the sentence accordingly to correctly refer to "the mean and the 95th percentile."

**Line 197**: '. . . slightly shorter events (Fig. 2d,e).'

We thank the reviewer for this suggestion. We have revised the sentence accordingly to correctly refer to ". . . slightly shorter events (Fig. 2d,e)."

**Line 194-200**: This paragraph is intended to explain the models' blocking duration, as introduced at the outset. However, the subsequent sentences conflate duration, frequency, and size. I recommend guiding the reader systematically through these three diagnostics and then, in a separate paragraph, explaining the biases by drawing on the contributions of the blocking characteristics.

We thank the reviewer for this important clarification and agree that the original paragraph conflated the description of blocking duration, frequency, and size, which reduced clarity. In the revised manuscript, we have restructured this section to first describe each blocking diagnostic separately (event number, duration, and size) in a systematic manner. We then discuss the implications of these diagnostics for blocking frequency biases in a separate synthesis paragraph. This reorganization improves readability and avoids mixing descriptive results with interpretation.

**Figure 1, caption**: Subplot labels are not correct. Please replace d), e) and g) by c), d) and e).

We thank the reviewer for pointing this out. We have corrected the subplot labels in the caption accordingly

(replacing d), e) and g) with c), d) and e)).

**Line 216**: The models are not multidecadal. The authors have to be more concise and may replace this part by '...in the multidecadal simulations/experiments of both storm-resolving models.'

We thank the reviewer for this clarification and agree that the original wording was imprecise. We have revised the text to replace "multidecadal storm-resolving models" with "multidecadal simulations of both storm-resolving models," which more accurately reflects the nature of the experiments.

**Line 237**: Do you refer to Table 4?

We thank the reviewer for pointing this out. Yes, this sentence refers to Table 4, and we have corrected the reference accordingly (replacing "Table 6" with "Table 4").

**Line 248**: 'Negative SST bias...'

We thank the reviewer for this suggestion. We have revised the wording accordingly, replacing "negative SST anomalies" with "negative SST bias" to improve precision and consistency.

**Line 270**: '...during wintertime.'

We thank the reviewer for this suggestion. We have revised the wording accordingly, replacing "...during DJF." with "...during wintertime.".

**Line 271**: ,"Over the North Atlantic,..."

We thank the reviewer for this suggestion. We have revised the wording accordingly, replacing "In the Atlantic," with "Over the North Atlantic,".

**Line 291**: The areas on which the averages are performed, do not really fit to the terminology of 'North Atlantic' and 'North Pacific'. I would suggest to adjust and reduce the size of the areas to the ocean basins to avoid misunderstandings with the terminology.

This comment is similar to the minor comment 3 from Reviewer 1. Here, I describe again our adjustments. The reviewer is correct that the North Atlantic and North Pacific domains used in our analysis extend over both oceanic and adjacent land regions and therefore encompass large longitudinal sectors of the Northern Hemisphere, rather than being limited strictly to the ocean basins. This domain definition was chosen to ensure capturing blocking regimes such as Ural blocking, which is especially prominent during boreal summer. In response, we have revised the domain definitions and reduced their spatial extent to better align with the ocean basins, thereby avoiding ambiguity in the terminology. The updated North Atlantic and North Pacific domains are now more consistent with common basin-based definitions and no longer extend excessively over adjacent continental region. We now show blocking characteristics for more standard Euro-Atlantic and North Pacific domains (e.g., Schiemann et al., 2020). In general, the results remain similar, however, we adapt where necessary. New proposed paragraph:
"Blocking statistics are evaluated over two broad longitudinal sectors of the Northern Hemisphere, referred to here as the North Atlantic (ATL) and North Pacific (PAC) regions. These domains are defined following Schiemann et al. (2020) and encompass both oceanic and adjacent continental areas, rather than being restricted strictly to the ocean basins. This choice allows for a consistent comparison with previous blocking climatologies and ensures that continental blocking regimes, such as Ural blocking, are adequately represented, particularly during boreal summer. The exact longitudinal and latitudinal bounds of each domain are for ATL 50–90 N, -90–90 E and PAC 40–90 N, 120–240 E ."

**Line 294/295+ 303/304**: To my understanding, the conclusions about the performance of the ICON simulation does not agree in these two sentences. While the authors emphasize that ICON has a small blocking frequency bias (RMSE=0.33) (line 294/295), they describe an underestimation of the number of blocking events by referring back to the Atlantic frequency bias. Please clarify if the presentation of the

results are correct. If so, I would recommend to revise the paragraphs with a more structured presentation of the results.

We thank the reviewer for identifying this apparent inconsistency. The results are correct, but we agree that the original wording did not sufficiently distinguish between the different blocking diagnostics. In summer over the North Atlantic, ICON hist indeed exhibits the smallest basin-averaged blocking frequency bias, as quantified by the RMSE. This indicates that the magnitude of spatial blocking-frequency errors relative to ERA5 is comparatively small across the domain. However, blocking frequency and blocking event count diagnose different aspects of blocking behavior. The event-based analysis shows that ICON hist underestimates the number of distinct blocking events, even though the resulting spatial frequency bias remains small. This implies that ICON tends to produce fewer blocking episodes that nevertheless occupy a sufficiently large spatial extent or persist long enough to yield near-correct blocking frequencies at many grid points. Thus, the low RMSE does not indicate accurate blocking initiation statistics, but rather reflects modest spatial frequency errors despite an underestimation of event counts. To avoid confusion, we have revised the Results section to clearly separate basin-averaged frequency metrics from event-based diagnostics and to explicitly explain how these quantities are related within the applied blocking methodology.

**Line 297**: "...more closely.." If the authors refer with these findings to Fig. 6a,b, I would not conclude that IFS AMIP more closely matches ERA5. There are still significant differences with a similar magnitude compared to IFS hist over the North Pacific. The extent is only somewhat smaller in IFS AMIP. I would recommend a more concise presentation of the differences between the models with regard to blocking characteristics.

We thank the reviewer for this careful observation and agree that the original phrasing overstated the degree of improvement in the North Pacific. While IFS AMIP exhibits a modest reduction in the spatial extent of blocking frequency biases compared to IFS hist, significant differences relative to ERA5 remain. In the revised manuscript, we have softened the wording to avoid implying close agreement with ERA5 and now describe the differences between the models more cautiously, emphasizing relative rather than absolute improvements.

**Line: 301**: I would assume the authors only refer to IFS hist and not to both IFS simulations here. 'In the North Atlantic, IFS hist produces . . . '

We agree with the reviewer. The sentence has been corrected to refer specifically to IFS hist rather than to both IFS simulations.

**Line 316**: 'The AMIP simulation shows larger blocks in general, i.e. the entire distribution is shifted towards an overestimation,. . . ' I would generally suggest to more concisely describe the distribution differences and highlight that not only the mean or median is different, but also the entire distribution is shifted relative to ERA5.

We thank the reviewer for this helpful suggestion. We note that this result changed after revising the basin definitions (from 90–270°E to 120–240°E). In the original domains, the IFS AMIP North Pacific blocking size distribution was shifted toward larger values relative to ERA5. In the revised domains, however, IFS AMIP now shows a systematic underestimation, with the mean, median, and the entire distribution shifted toward smaller block sizes compared to ERA5. In the revised manuscript, we have updated the description accordingly and now explicitly state whether the full distribution is shifted, rather than referring only to mean or median differences.

**Line 320-321**: ICON hist and not IFS AMIP has the lowest RMSE for blocking frequency in the North Atlantic according to Table 5. Also, what is meant by the term "balanced performance"? Do we observe a compensation of biases between different blocking characteristics leading to an overall low RMSE? Please clarify the contribution of all three blocking characteristics.

We thank the reviewer for identifying this inconsistency. Indeed, ICON hist has the lowest RMSE for

blocking frequency in the North Atlantic according to Table 5, and this has now been corrected in the revised manuscript. IFS AMIP is the second-best performer and is now described accordingly. We also agree that the term "balanced performance", which referred to IFS AMIP, was ambiguous. What we intended to convey is that in IFS AMIP different blocking characteristics partly compensate each other: the model slightly underestimates the number of blocking events, but this is offset by somewhat larger and longer-lived blocks, such that the basin-averaged blocking frequency is relatively close to ERA5. In the revised manuscript, we now explicitly describe the contributions from event number, duration, and size instead of using the vague term "balanced performance."

**Line 325**: ICON does not simulate smaller blocks in the Pacific according to Fig. 7f.

We thank the reviewer for pointing this out. They are correct: according to Fig. 7f, ICON hist produces larger blocks in the North Pacific rather than smaller ones. We have corrected the text accordingly in the revised manuscript.

**Line 326-327**: ICON-hist and not IFS AMIP has the highest block count over the North Pacific according to Fig. 7d.

We thank the reviewer for identifying this error. The text has been corrected to replace "IFS AMIP" with "ICON hist", which indeed shows the highest block count over the North Pacific in Fig. 7d.

**Line 332**: "... than the coarse-resolution CMIP models."

We thank the reviewer for this suggestion. The wording has been revised to read: However, their performance is not consistently better than the coarse-resolution CMIP models.

**Fig. 7d**: Where is the line for the mean of IFS amip?

We thank the reviewer for pointing this out. In Fig. 7d (from the revised manuscript), the mean and median number of North Pacific blocking events in IFS AMIP are very close to ERA5, whereas the upper tail (95th percentile) is overestimated. We have clarified this in the revised text by explicitly describing the mean, median, and upper-percentile behavior, rather than referring only to a general under- or overestimation in lines 302-303 of the original manuscript.

**Figure 7, caption**: '...and their properties'. Providing the explicit list of properties/characteristics with subplot references would be helpful here, e.g., count (a,d), duration (b,e), size (c,f)

We thank the reviewer for this helpful suggestion. We have revised the caption of Fig. 7 to explicitly list the blocking properties and their corresponding subplots (count: a,d; duration: b,e; size: c,f), improving clarity and readability.

**Line 335**: '...midlatitudes.' The authors need to provide a reference to a figure of the their study or cite a previous study.

We thank the reviewer for pointing this out. We have added an appropriate reference to support this statement, documenting the seasonal differences in jet structure between winter and summer..

**Line 339-340**: IFS hist shows a positive U bias here as well (Fig. 8a).

We thank the reviewer for this clarification. They are correct that IFS hist also exhibits a positive zonal wind bias southeast of Greenland. In the revised manuscript, we have corrected the text accordingly.

**Line 346**: "...more intense jets..." Do you refer to multiple jets here? Subtropical and polar jet? Please clarify.

We thank the reviewer for this important clarification. The analysis is based on seasonal-mean zonal wind at 500 hPa, which does not allow for a clear separation between the subtropical and eddy-driven jets,

particularly in boreal summer when their latitudinal separation is reduced and the jets may partially merge. In the revised manuscript, we therefore avoid referring to multiple jet structures and instead describe the strength and position of the seasonal-mean zonal-wind maximum. The text has been revised accordingly to improve clarity and precision.

**Line 347/348**: To my understanding, CMIP indicates a slight equatorward shift rather than a broader jet over Eurasia.

We thank the reviewer for this comment. We agree that the CMIP6 ensemble mean has not been properly described. In the revised manuscript, we clarify that by "broader jet" we refer to a redistribution of zonal winds in the seasonal-mean sense, characterized by reduced winds near the climatological jet core and enhanced winds on its equatorward and poleward flanks, rather than a simple latitudinal displacement of the jet axis. We now describe the CMIP6 bias more explicitly and note that its weaker and less coherent pattern partly reflects ensemble averaging across models with differing jet structures. The text has been revised accordingly to improve clarity and avoid ambiguity.

**Line 348/349**: The bias in CMIP is generally lower than in IFS AMIP, however, the RMSE for IFS amip is generally lower than for CMIP. Does this inconsistency emerges due to a stronger cancellation of biases with opposite signs in IFS amip and thus the limitation of the large size of areas, upon which the averaged RMSE values are calculated on?

We thank the reviewer for this insightful comment. The apparent inconsistency does not arise from cancellation of biases with opposite signs, since the RMSE is based on squared differences, but rather from differences in the spatial extent of the biases. In IFS AMIP, zonal-wind biases tend to be larger in amplitude but are confined to relatively narrow latitude bands and localized regions. As a result, fewer grid points contribute substantially to the basin-averaged RMSE. In contrast, the CMIP6 ensemble mean exhibits weaker local biases, but these biases are spatially more widespread across the basin, affecting a larger fraction of grid points. This broader spatial coverage leads to a higher basin-averaged RMSE despite smaller local amplitudes. We have clarified this interpretation in the revised manuscript and emphasize that basin-averaged RMSE values should be interpreted alongside the spatial structure of the bias patterns shown in the maps.

**Line 353**: The overestimation of blocking largely occurs over North America and thus downstream of the poleward jet shift. Thus, it would not contradict the common inverse relationship between jet intensity and blocking occurrence.

We thank the reviewer for this clarification. We agree that a substantial part of the Pacific blocking overestimation occurs downstream of the poleward jet shift over North America, which is consistent with the commonly reported inverse relationship between jet intensity and blocking occurrence. We have revised the text accordingly. At the same time, we note that enhanced blocking is also present in regions such as the Bering Sea, where it coincides with stronger-than-observed westerlies. We now emphasize that the jet–blocking relationship in the Pacific is regionally dependent rather than uniform.

**Line 369**: 'negative SST bias'

We thank the reviewer for this suggestion. We have modified "negative SST anomalies" to "negative SST bias".

**Line 375**: 'warm SST bias'

We thank the reviewer for this suggestion. We have modified "warm SST anomalies" to "warm SST bias".

**Line 376**: "Such positive bias" We thank the reviewer for this suggestion. We have modified "Such positive anomalies" to "Such positive bias"

**Line 380**: "the impact of this cold bias"

We thank the reviewer for this suggestion. We have modified "the impact of this bias" to "the impact of this cold bias"

**Line 386**: "...when biases are strong.." - Why is the influence of SST anomalies on blocking only present when biases are strong? Bias differences in SST and blocking just indicate that a reduced SST bias might be related to a reduced blocking bias. Also what is meant by 'spatially structured'? I would recommend to remove the part after the comma.

We agree with the reviewer. The phrasing "particularly when biases are strong and spatially structured" was unnecessary and potentially misleading. We have removed this qualifier in the revised manuscript and now state the relationship more generally.

**Figure 9, caption**: (c) instead of (d)

We thank the reviewer for noticing this. We have modified "(d)" to "(c)".

**Line 292**: Why is the acronym (IFS hist) introduced here again?

We thank the reviewer for pointing this out. The acronym IFS hist was unnecessarily reintroduced at this point in the manuscript. We have removed the redundant definition in the revised version.

**Line 396**: ICON simulations feature a weaker storm track compared to IFS AMIP or ERA5?

We thank the reviewer for this clarification. The weaker storm-track in ICON refers to a reduction relative to both ERA5 and IFS AMIP. We have revised the text to state this explicitly in the manuscript.

**Line 403**: What is meant by "meridionally spread"? This needs to be better expressed, for instance: "the jet is not exclusively tied to the mid-latitudes, and rather present on a wider latitude range" ... do you mean something like that?

We thank the reviewer for this clarification. By "meridionally spread" we meant that, in summer, storm-tracks and the associated jet are distributed over a wider latitudinal range and are not as tightly confined to the midlatitudes as in winter. We have revised the wording accordingly to make this interpretation explicit.

**Line 406-416 + Fig. 11**: There are a couple of shortcomings related to Fig. 11 and its description, as mentioned in major comment 5. Why does the figure illustrates only the relationship of blocking and storm tracks? Why is the relationship in this framework only shown for summer and not for winter in the previous section 4.1? The results are not really convincing as the relationship is relatively weak. Is this again a consequence of averaging over too large domain (smoothing and/or cancellation of signals)? I would recommend to revise the figure accordingly or consider removing it or move it to SI.

We thank the reviewer for this detailed comment. We agree that Fig. 11 and its associated discussion were not sufficiently well motivated and that the diagnosed relationships are relatively weak, particularly when evaluated over the revised basin definitions. As the reviewer suggests, this likely reflects a combination of spatial averaging over large domains and partial cancellation of regional signals. Figure 11 was originally shown only for summer because the corresponding wintertime correlations were even weaker and less spatially coherent. However, we acknowledge that this choice was not sufficiently justified and introduced an inconsistency between Sections 4.1 and 4.2. To address these concerns, we have moved Fig. 11 to the Supplementary Material, where we now also include the corresponding wintertime analysis for completeness. The discussion in the main text has been revised to avoid overemphasizing this diagnostic and to maintain a consistent analytical framework between winter and summer. The revised presentation reflects the exploratory nature of this analysis and clarifies that storm-track intensity alone provides only limited explanatory influence for blocking biases, particularly in storm-resolving configurations.

**Figure 9+10**: There is no hatching visible, eben though differences are quite large regionally. Please double-check if there is really no significance.

We thank the reviewer for pointing this out. In Figs. 9–10, the hatching does not indicate statistical significance. Instead, it highlights grid points where the relative difference with respect to ERA5 exceeds 80%, as stated in the captions. For the SST fields (Fig. 9), relative differences can exceed 80% mainly at high latitudes because the absolute values are small, so even modest absolute biases can translate into large relative differences. In contrast, for the storm-tracks fields (Fig. 10), typical ERA5 values are on the order of 30–40 m, whereas biases are generally $\lesssim$10 m. Thus, even when the absolute biases are dynamically meaningful, the relative differences rarely exceed the 80% threshold, and hatching is therefore sparse or absent. We have nevertheless rechecked the plotting and confirmed that the hatching criterion is applied consistently in the revised manuscript.

**Figure 11, caption**: What is meant by 'selected simulations'?

We thank the reviewer for pointing this out. By "selected simulations", we meant that the analysis is restricted to the historical simulations only, and therefore does not include the SSP3-7.0 future-climate experiments. All historical CMIP6 models and all historical storm-resolving simulations analysed in this study are included. We have clarified this, in figures S7 and S8 (before Fig. 11).

**Line 420/421+ line 443 +448**: The authors regularly point out that substantial biases must be taken into account when assessing blocking trends. As mentioned in the major comment 3, I would thus recommend to remove section 4.3 from the study.

We thank the reviewer for this important comment and fully agree that the assessment of future blocking changes is complicated by the presence of substantial present-day biases in both storm-resolving and conventional climate models. We have explicitly acknowledged this limitation throughout the manuscript to avoid overstating the robustness of projected trends. We also agree that a comprehensive and fully process-based analysis of blocking trends would require a more dedicated treatment and is beyond the scope of the present study. Nevertheless, we believe that it remains valuable to provide a clearly qualified, exploratory assessment of how blocking characteristics evolve in the storm-resolving simulations analysed here, particularly where the projected signals are consistent with those reported in previous CMIP-based studies. As we discussed in major comment 3, conventional climate models also exhibit persistent blocking biases, yet they continue to underpin much of our current understanding of future circulation changes. In this context, documenting similarities and differences between storm-resolving simulations and CMIP projections provides useful information for the community and helps place the storm-resolving results within the broader literature. In response to the reviewer's concern, we have substantially revised Section 4.3. The section has been condensed, its scope reduced, and all causal or overconfident interpretations have been removed. The revised text now focuses on a descriptive comparison of projected changes, explicitly highlights the limitations imposed by present-day biases, and frames the results as first insights rather than definitive projections. We further emphasize that a more detailed and process-oriented assessment of blocking trends in storm-resolving models is a natural subject for future work.

**Line 431**: The suggested equatorward shift is not confirmed by a negative pattern directly northward of the positive pattern. The negative pattern appears over Eurasia (Fig. 12a).

We thank the reviewer for this helpful clarification. We agree that the projected winter blocking changes do not exhibit a clear meridional dipole structure that would robustly indicate an equatorward shift of North Pacific blocking. In particular, the strongest decrease occurs over northern Eurasia, while the increase in blocking frequency is located south of the climatological North Pacific blocking maximum rather than being paired with a local decrease immediately poleward. In the revised manuscript, we have therefore removed the statement referring to an "equatorward shift" and now describe the pattern more accurately as a redistribution of winter blocking activity, characterized by reduced high-latitude blocking over Eurasia and enhanced blocking at lower latitudes in the North Pacific. This revised wording better reflects the spatial structure shown in Fig. 11a (before Fig. 12a) and avoids overinterpretation of the projected changes.

**Line 435**: Fig. 13d clearly shows the typical North Atlantic warming hole. So the reduced blocking does not align with a North Atlantic warming pattern.

We thank the reviewer for this careful observation and agree with the assessment. As also noted by Reviewer 1, the reduced blocking shown in Fig. 12b is co-located with the North Atlantic warming hole evident in Fig. 13d, rather than with a basin-wide warming of the North Atlantic. This spatial structure complicates any interpretation based on a simple thermodynamic link between warmer SSTs and reduced blocking in this region. In the original manuscript, our interpretation overstated the robustness of such a link. In the revised version, we therefore restrict the Results section to a factual description of the spatial correspondence between SST anomalies and blocking changes, without attributing causality. In the Discussion, we note more cautiously that SST anomalies may still influence blocking regionally (particularly along the climatological storm-track corridor where diabatic processes and air–sea fluxes are most active) but emphasize that this influence is neither uniform across the basin nor sufficient on its own to explain the blocking response.

**Line 439-442**: The results on the number of events (count) should be presented here as well (Fig. 12c,f,i,l).

We thank the reviewer for this helpful suggestion. We agree that changes in blocking count should be reported alongside duration and size to provide a complete description of projected changes in blocking characteristics (Fig. 12c,f,i,l). In the revised manuscript, we now explicitly describe projected changes in blocking count for both basins and seasons. Consistent with the reviewer's observation, these changes are generally modest and often not statistically significant, but they show contrasting seasonal tendencies: little change in winter over the Atlantic, a decrease in summer Atlantic blocking events, a slight decrease in winter Pacific blocking count, and a modest increase in summer Pacific blocking events. These additions are presented descriptively, in line with the exploratory nature of Section 4.3 and without overstating their robustness.

**Line 443-455**: Even though there is a small reference to it earlier in the text, this paragraphs lacks a more detailed discussion on the results related to the SST trend (Fig. 13c,d).

We thank the reviewer for this comment. We agree that the original paragraph did not sufficiently relate the projected circulation changes to the SST trends shown in Fig. 13c,d. In the revised manuscript, we now explicitly describe the main SST patterns under SSP3-7.0 and discuss how they spatially coincide with projected jet and blocking changes, while avoiding strong causal claims. In particular, we now note the persistence of a North Atlantic warming hole, contrasted with enhanced warming along the Gulf Stream region, and describe how these SST patterns co-occur with changes in jet strength and blocking frequency. For the North Pacific, we describe the basin-wide warming and its meridional structure, especially the stronger midlatitude warming in winter. Consistent with the overall cautious framing of Section 4.3, these SST-related aspects are presented descriptively, with any mechanistic interpretation deferred to the Discussion section.

**Line 489-507 + Line 527-536**: These paragraphs are an extension of the results rather than a discussion of your results in the context of other studies. These paragraphs need to focus more on how the results relate to current research and should be shortened for clarity, as mentioned major comment 4.

We thank the reviewer for this important comment. We agree that parts of the original Discussion were overly descriptive and too closely resembled an extension of the Results section, rather than placing the findings in the context of existing research. In response, we have substantially revised and restructured the Discussion (in line with the major comment 4 and Reviewer 1's related comment). The section is now organized into clearly defined thematic subsections, namely: (i) Blocking Representation in Storm-Resolving Models, (ii) Role of SST Biases and Air–Sea Coupling, (iii) Jet, Storm-Track, and Blocking Interactions, (iv) Influence of Moist Processes, (v) Implications for Future Blocking Changes and (vi) Final Remarks. This structure allows us to explicitly relate our results to previous studies in each thematic area, while avoiding repetition of results already presented in Sections 4.1–4.3. In addition, the text has been condensed throughout, and descriptive passages have been shortened in favor of interpretation and comparison with existing literature. The climate-change subsection has been reframed to emphasize its exploratory nature and to clearly acknowledge the limitations imposed by present-day model biases. We believe these changes significantly improve clarity, focus, and alignment with the current research state, as recommended by the reviewer.

**Line 515**: 'These remaining biases could be due to the misrepresentation of moist processes...'

We thank the reviewer for pointing this out: We have modified from "These remaining biases could be due to the representation of moist processes" to "These remaining biases could be due to the misrepresentation of moist processes"

**Line 541-553**: None of the raised bullet points include a reference to blocking under climate change. Instead, the most important future research branch motivated by the present study is the improvement of the representation of blocking in multi-decadal historical climate simulations to facilitate a more reliable estimate on how blocking will change regionally and seasonally under climate change conditions.

We thank the reviewer for this important suggestion. We agree that a key future research priority motivated by this study is the improvement of blocking representation in multi-decadal historical simulations, as this is essential for obtaining more reliable estimates of how blocking may change regionally and seasonally under climate change. In response, we have added an explicit bullet point to the list of future research directions emphasizing the need to reduce present-day blocking biases in storm-resolving and coupled climate models as a prerequisite for robust climate change assessments. This addition aligns with the cautious interpretation of future blocking changes adopted throughout the revised manuscript.

**Line 571 + 574/575**: This key finding indicate that storm-resolving models still have blocking biases and do not advance blocking simulation. As mentioned in the major comment 1, I would recommend revising the title of the study.

We agree with the reviewer and in response to this comment and major comment 1, the revised title reads: "Atmospheric Blocking Representation in Storm-Resolving Climate Models under Historical and Future Forcing."

**Line 622**: Even though the authors mention the assistance of Large-Language Models (LLM) such as ChatGPT, I would suggest being careful to include longer sentence structures. Sentences in which an insertion is included separated by long dashes (em-dashes) is a typical output feature by ChatGPT.

We thank the reviewer for this observation. ChatGPT was used for grammar and spelling checks. In the revised manuscript, we have carefully reviewed the text and revised sentence structure where necessary, avoiding overly long or complex constructions and reducing the use of em dashes to improve clarity and readability.

**References**

Brunner, L., Poschlod, B., Dutra, E., Fischer, E. M., Martius, O., and Sillmann, J. (2025). A global perspective on the spatial representation of climate extremes from km-scale models. *Environmental Research Letters*, 20(7):074054.

Dolores-Tesillos, E., Martius, O., and Quinting, J. (2025). On the role of moist and dry processes in atmospheric blocking biases in the euro-atlantic region in cmip6. *Weather and Climate Dynamics*, 6(2):471–487.

Hohenegger, C., Korn, P., Linardakis, L., Redler, R., Schnur, R., Adamidis, P., Bao, J., Bastin, S., Behravesh, M., Bergemann, M., Biercamp, J., Bockelmann, H., Brokopf, R., Brüggemann, N., Casaroli, L., Chegini, F., Datseris, G., Esch, M., George, G., Giorgetta, M., Gutjahr, O., Haak, H., Hanke, M., Ilyina, T., Jahns, T., Jungclaus, J., Kern, M., Klocke, D., Kluft, L., Kölling, T., Kornblueh, L., Kosukhin, S., Kroll, C., Lee, J., Mauritsen, T., Mehlmann, C., Mieslinger, T., Naumann, A. K., Paccini, L., Peinado, A., Praturi, D. S., Putrasahan, D., Rast, S., Riddick, T., Roeber, N., Schmidt, H., Schulzweida, U., Schütte, F., Segura, H., Shevchenko, R., Singh, V., Specht, M., Stephan, C. C., von Storch, J.-S., Vogel, R., Wengel, C., Winkler, M., Ziemen, F., Marotzke, J., and Stevens, B. (2023). ICON-Sapphire: simulating the components of the Earth system and their interactions at kilometer and subkilometer scales. *Geoscientific Model Development*, 16(2):779–811.

Palmer, T. E., McSweeney, C. F., Booth, B. B., Priestley, M. D., Davini, P., Brunner, L., Borchert, L., and Menary, M. B. (2022). Performance based sub-selection of cmip6 models for impact assessments in europe. *Earth System Dynamics Discussions*, 2022:1–45.

Poujol, B., Lee, J., Rackow, T., Rotach, M. W., and Ban, N. (2025). Are the largest benefits of kilometer-scale climate models over mountains or over flatland? *Geophysical Research Letters*. First published: 28 April 2025. Correction published: 28 July 2025.

Prein, A. F., Langhans, W., Fosser, G., Ferrone, A., Ban, N., Goergen, K., Keller, M., Tölle, M., Gutjahr, O., Feser, F., Brisson, E., Kollet, S., Schmidli, J., van Lipzig, N. P. M., and Leung, R. (2015). A review on regional convection-permitting climate modeling: Demonstrations, prospects, and challenges. *Reviews of Geophysics*, 53(2):323–361.

Quinting, J. F. and Grams, C. M. (2022). Eulerian identification of ascending airstreams (elias 2.0) in numerical weather prediction and climate models–part 1: Development of deep learning model. *Geoscientific Model Development*, 15(2):715–730.

Schiemann, R., Athanasiadis, P., Barriopedro, D., Doblas-Reyes, F., Lohmann, K., Roberts, M. J., Sein, D. V., Roberts, C. D., Terray, L., and Vidale, P. L. (2020). Northern hemisphere blocking simulation in current climate models: evaluating progress from the climate model intercomparison project phase 5 to 6 and sensitivity to resolution. *Weather and Climate Dynamics*, 1(1):277–292.

Tibaldi, S. and Molteni, F. (1990). On the operational predictability of blocking. *Tellus A*, 42(3):343–365.

Wille, J. D., Koch, R., Becker, T., and Fischer, E. (2025). Extreme precipitation depiction in convection-permitting earth system models within the nextgems project. *Journal of Advances in Modeling Earth Systems*. First published: 14 July 2025.